# Rainfall intensification increases the contribution of rewetting pulses to soil heterotrophic respiration

Stefano Manzoni[1,2], Arjun Chakrawal[1,2], Thomas Fischer[3], Joshua P. Schimel[4], Amilcare Porporato[5], Giulia Vico[6]

[1]Department of Physical Geography, Stockholm University, 10691 Stockholm, Sweden
[2]Bolin Centre for Climate Research, 10691 Stockholm, Sweden
[3]Central Analytical Laboratory, Brandenburg University of Technology, Cottbus, Germany
[4]Department of Ecology, Evolution, and Marine Biology, University of California, Santa Barbara, USA
[5]Department of Civil and environmental Engineering, Princeton University, Princeton, USA
[6]Department of Crop Production Ecology, Swedish University of Agricultural Sciences, Uppsala, Sweden

*Correspondence to*: Stefano Manzoni (stefano.manzoni@natgeo.su.se)

**Abstract.** Soil drying and wetting cycles promote carbon (C) release through large heterotrophic respiration pulses at rewetting, known as 'Birch' effect. Empirical evidence shows that drier conditions before rewetting and larger changes in soil moisture at rewetting cause larger respiration pulses. Because soil moisture varies in response to rainfall, also these respiration pulses depend on the random timing and intensity of precipitation. In addition to rewetting pulses, heterotrophic respiration continues during soil drying, eventually ceasing when soils are too dry to sustain microbial activity. The importance of respiration pulses in contributing to the overall soil heterotrophic respiration flux has been demonstrated empirically, but no theoretical investigation has so far evaluated how the relative contribution of these pulses may change along climatic gradients or as precipitation regimes shift in a given location. To fill this gap, we start by assuming that heterotrophic respiration rates during soil drying and pulses at rewetting can be treated as random variables dependent on soil moisture fluctuations, and develop a stochastic model for soil heterotrophic respiration rates that analytically links the statistical properties of respiration to those of precipitation. Model results show that both the mean rewetting pulse respiration and the mean respiration during drying increase with increasing mean precipitation. However, the contribution of respiration pulses to the total heterotrophic respiration increases with decreasing precipitation frequency and to a lesser degree with decreasing precipitation depth, leading to an overall higher contribution of respiration pulses under future more intermittent and intense precipitation. Specifically, higher rainfall intermittency at constant total rainfall can increase the contribution of respiration pulses up to ~10 or 20% of the total heterotrophic respiration in mineral and organic soils, respectively. Moreover, the variability of both components of soil heterotrophic respiration is also predicted to increase under these conditions. Therefore, with future more intermittent precipitation, respiration pulses and the associated nutrient release will intensify and become more variable, contributing more to soil biogeochemical cycling.

## 1. Introduction

Heterotrophic respiration pulses often occur after dry soils are wetted by rainfall or irrigation (Barnard et al., 2020; Borken and Matzner, 2009; Canarini et al., 2017; Jarvis et al., 2007; Kim et al., 2012). The respiration rates achieved at rewetting can be much higher than the rates maintained under permanently moist conditions, suggesting that the rewetting itself triggers a disproportionly high $CO_2$ production. Even if they are short-lived, these pulses can contribute a significant amount of the annual $CO_2$ release (Kim et al., 2012; Li et al., 2004; Yan et al., 2014). Their occurrence had been documented as long ago as Birch (1958)—for which the phenomenon has been named the 'Birch effect'—but they remain difficult to explain and predict. Respiration pulses are larger when the change in soil moisture is larger and when the soil was drier before rewetting, as shown by observations under both laboratory (Birch, 1958; Fischer, 2009; Guo et al., 2014; Lado-Monserrat et al., 2014; Schaeffer et al., 2017; Williams and Xia, 2009) and field conditions (Cable et al., 2008; Carbone et al., 2011; Lopez-Ballesteros et al.,

2016; Rubio and Detto, 2017; Unger et al., 2010; Yan et al., 2014). Besides $CO_2$ displacement at rewetting, several mechanisms linked to microbial processes have been postulated to explain these patterns (Barnard et al., 2020; Canarini et al., 2017; Kim et al., 2012; Schimel et al., 2007). It has been argued that cell lysis due to a rapid increase in water potential and subsequent consumption of the dead cells may cause the pulse (Bottner, 1985). Later measurements showed that little cell lysis occurs,
but that intracellular materials (osmolytes) can be released at rewetting, contributing to the respiration pulse (Fierer and Schimel, 2003). However, in some soils microbial cells become dormant during drying rather than accumulating osmolytes (Boot et al., 2013). It is thus possible that respiration pulses are triggered by a physical process associated with the rewetting event—possibly re-establishment of hydrologic connectivity between substrates and microorganisms (Manzoni et al., 2016), or physical disruption of soil aggregates releasing old organic matter (Homyak et al., 2018). Indeed, there is a strong correlation
between the $CO_2$ production after rewetting and the amount of extractable organic C consumed, suggesting that extractable C accumulated during the previous dry period could fuel the respiration pulse (Canarini et al., 2017; Guo et al., 2014; Williams and Xia, 2009). It is likely that multiple mechanisms work in concert, shifting their relative importance under different conditions (Slessarev and Schimel, 2020).

The focus on the processes causing respiration pulses resulted in extensive work conducted under idealized laboratory
conditions, in which soil moisture changes were controlled, typically following a regular pattern of drying and wetting (Fierer and Schimel, 2002; Miller et al., 2005; Shi and Marschner, 2014, 2015; Xiang et al., 2008). However, soil moisture varies randomly due to the stochastic nature of rainfall events (Katul et al., 2007; Rodriguez-Iturbe and Porporato, 2004), and this temporal variability can either promote or decrease soil organic C storage depending on its effects on soil microbes (Lehmann et al., 2020). Two features of soil moisture dynamics are particularly important because they directly affect the intensity of a
respiration pulse—the duration of dry periods and the soil moisture increment at rewetting. Therefore, experimental designs based on regular cycles of drying and wetting do not allow exploring how the stochastic nature of soil moisture fluctuations may affect respiration pulses. Capturing the effect of these stochastic fluctuations can be important as climatic changes are altering rainfall patterns—often lengthening the duration of droughts and increasing the intensity of the (less frequent) rainfall events (IPCC, 2012).

To quantify how the long-term mean heterotrophic respiration varies as a function of rainfall statistical properties (duration of dry periods and intensity), we developed a stochastic soil moisture and respiration model, parameterized using available respiration data. Specifically, we ask—how does variability in rainfall translate in variability in respiration pulses? How does the long-term mean contribution of respiration pulses vary along climatic gradients? These questions are motivated by the hypothesis that respiration pulses contribute a larger proportion of soil heterotrophic respiration under climates with more
intermittent and intense rainfall events, compared to climates in which soil moisture variations are mild. If that is the case, future climatic conditions characterized by longer droughts and more intense rainfall events are expected to increase the overall role of respiration pulses in ecosystem C budgets.

## 2.    Methods

### 2.1.    Theory

The theoretical framework is illustrated in Figure 1. We start from the premise that heterotrophic respiration follows changes in soil moisture during drying ($R_d$) and that respiration pulses occur immediately following rewetting. As such, respiration pulses depend on both the soil moisture at the end of the dry period and the soil moisture increase caused by rainfall ($R_r$). The stochasticity of rainfall timing and amount determines a range of possible durations of dry spells and soil moisture increments when rainfall occurs. As a result, respiration can be regarded as a stochastic process. To characterize statistically the two types
of respiration, the statistical properties of both soil moisture and soil moisture changes at rewetting are needed. These statistical

properties are included in the probability density function (PDF) of soil moisture, and the joint PDF of soil moisture and its increase at rewetting. Both distributions are derived in Section 2.1.1. The PDF of respiration rates during drying and respiration pulses at rewetting are derived in Sections 2.1.2 and 2.1.3, respectively. All symbols are defined in Table 1.

### 2.1.1. Soil moisture dynamics

Soil moisture varies in response to rainfall events and the subsequent loss of soil water by deep infiltration below the rooting zone and evapotranspiration. The dynamics of soil moisture in the rooting zone (the most biogeochemically active soil layer) can be described by the mass balance equation (Laio et al., 2001; Rodriguez-Iturbe and Porporato, 2004),

$$nZ_r \frac{ds}{dt} = P(t) - E(s(t)) - L(s(t), t), \tag{1}$$

where $s$ is the saturation level (i.e., the relative volumetric soil moisture), $n$ is the soil porosity, $Z_r$ is the rooting depth, and $P$, $E$, and $L$ represent precipitation inputs, evapotranspiration rate, and the combination of water losses due to percolation below the rooting zone and surface runoff. Equation (1) is interpreted at the daily time scale. Given our aim to describe the statistical properties of respiration rather than the details of soil moisture dynamics, we simplify the soil moisture mass balance equation to a form that is analytically tractable. Thus, we assume that evapotranspiration is the dominant water loss when soil moisture is lower than a threshold $s_1$ (equivalent to the soil field capacity), whereas runoff and deep percolation dominate above this threshold. Also, runoff and percolation are assumed to occur rapidly compared to the timescales of the soil dry-down (free drainage conditions), so that, after a precipitation event that brings soil moisture above the level $s_1$, soil moisture decreases instantaneously to $s_1$. For simplicity, evapotranspiration is modelled as a linear function of soil moisture (Porporato et al., 2004),

$$E = \frac{s - s_w}{s_1 - s_w} E_{max} = x E_{max}, \tag{2}$$

where $E_{max}$ is the maximum rate of evapotranspiration, $s_w$ is the plant wilting point (below which $ET$ becomes negligible), and $s_1$ is the threshold above which runoff and percolation are dominant. In the second equality, a normalized soil moisture denoted by $x$ is introduced to further simplify the notation. With these assumptions and definitions, $s$ ranges between $s_w$ and $s_1$, while the normalized soil moisture varies between 0 and 1.

Precipitation is treated as a marked Poisson process with mean frequency $\lambda$ and rain-event depths exponentially distributed with mean $\alpha$. At each rain event, soil moisture increases by an amount corresponding to the rain event depth (normalized by $nZ_r$), unless the depth exceeds the available soil storage capacity (i.e., $nZ_r(s_1 - s)$). Assuming that rainfall exceeding this capacity is routed to runoff, the PDF of soil moisture increments due to a rain event, $y$, for a given soil moisture at the end of the dry period, $x_d$, is given by (Laio et al., 2001)

$$p_y(y|x_d) = \theta[(1 - x_d) - y]\gamma e^{-\gamma y} + \delta[y - (1 - x_d)]e^{-\gamma(1 - x_d)}, \tag{3}$$

where $p_y(y|x_d)$ is the PDF of $y$ *conditional* on soil moisture at the end of the dry period, $x_d$; $\theta[\cdot]$ is the Heaviside step function; $\delta[\cdot]$ is the Dirac delta function; and $\gamma$ is a parameter group defined as $\gamma = \frac{nZ_r(s_1 - s_w)}{\alpha}$ ($\gamma$ can be interpreted as the number of average rainfall events needed to replenish the plant available soil water). The first term on the right-hand side of Eq. (3) represents the probability density of a soil moisture increase $y$ equal to the rainfall depth ($\theta[\cdot]$ is equal to one for $y<1-x_d$; zero otherwise). The second term represents the probability of a soil moisture increase from the value $x_d$ to the soil field capacity ($x=1$). This term is also referred to as an 'atom of probability' because $\delta[\cdot]$ is equal to zero for all soil moisture increments, except $y=1-x_d$, at which $\delta[\cdot] = \infty$.

With this stochastic description of precipitation events and further assuming stochastic stationary conditions, the PDF of the normalized soil moisture driven by the dynamics in Eq. (1) can be obtained analytically and reads (Porporato et al., 2004)

$$p_x(x) = C_1 \frac{e^{-x\gamma} x^{-1+\frac{\lambda}{\eta}}}{\eta}, \tag{4}$$

where $\eta$ is a parameter group defined as $\eta = \frac{E_{max}}{nZ_r(s_1-s_w)}$. $C_1$ is a normalization constant that guarantees that the area under $p_x(x)$ between $x=0$ and 1 is one,

$$C_1 = \frac{\eta\gamma^{\frac{\lambda}{\eta}}}{\Gamma\left[\frac{\lambda}{\eta}\right]-\Gamma\left[\frac{\lambda}{\eta}\gamma\right]}, \tag{5}$$

where $\Gamma[\cdot]$ and $\Gamma[\cdot,\cdot]$ are the complete and incomplete gamma functions (defined in Table 1). The PDF of soil moisture is the basis to obtain the PDF of respiration during soil drying (Section 2.1.2).

The last distribution needed to calculate the statistical properties of soil respiration pulses (Section 2.1.3) is the joint PDF of soil moisture at the end of a dry period and soil moisture increase due to precipitation events, denoted by $p_{y,x_d}(y,x_d)$ (note that both $y$ and $x_d$ are stochastic variables in this joint PDF). Thanks to the properties of the Poisson process, the PDF of soil moisture at the end of the dry period is equal to the PDF of soil moisture at a generic time (Cox and Miller, 2001); i.e., $p_{x_d}(x_d) = p_x(x)$. Because precipitation does not depend on antecedent soil moisture conditions in this model, the PDF of soil

moisture at the end of a dry period is independent of the PDF of the subsequent precipitation event and soil moisture increase. Thus, the joint PDF of $x_d$ and $y$ is given by the product of the PDFs of $x_d$ (Eq. (4)) and of $y$ conditional to $x_d$ (Eq. (3)),

$$p_{y,x_d}(y,x_d) = p_x(x_d)p_y(y|x_d). \tag{6}$$

### 2.1.2. Heterotrophic respiration during soil drying

During a dry period, the heterotrophic respiration rate decreases in response to the gradual decrease in soil moisture, following a concave-downward trend (Manzoni et al., 2012; Moyano et al., 2012). Consistent with the hydrologic model setup, we

assume that the soil drains rapidly and hence does not remain under saturated conditions long enough to develop anoxic conditions. It is thus reasonable to assume that respiration declines between the soil field capacity (equivalent to $s_1$ in this model) and a lower soil moisture threshold for microbial activity. This lower threshold corresponds to water potential levels around -15 MPa in sieved soil samples (Manzoni and Katul, 2014), but here we assume that respiration becomes much smaller than rates under well-watered conditions already at the plant wilting point $s_w$; i.e., at a water potential of -1.5 MPa. This

assumption is motivated by the observation that in intact soil cores and under field conditions respiration stops in wetter conditions than at -15 MPa (e.g., -2.7 MPa; Carbone et al., 2011). Moreover, this allows keeping the parameter number to a minimum, consistent with the minimal soil moisture balance model of Eq. (1) and (2) and the overall idealized representation of soil heterotrophic respiration. The respiration decrease with a lower threshold $s_w$ (corresponding to $x=0$) can be captured by a parabolic relation,

$$R_d = R_{d,max}(2x - x^2), \tag{7}$$

where $R_d$ denotes the respiration rate during drying and $R_{d,max}$ is the maximum respiration rate in the absence of rapid rewetting (i.e., $R_d$ at $x=1$ or $s=s_1$). Using other monotonic and concave-downward relations between respiration and soil moisture would not qualitatively alter the results.

In Eq. (1), soil moisture is a random variable, whose PDF follows Eq. (4). Therefore, $R_d$ from Eq. (7) is also a random variable, which can be obtained from the PDF of soil moisture using the derived distribution approach, also referred to as Jacobian rule

(Kottegoda and Rosso, 1998),

$$p_{R_d}(R_d) = p_x\big(x(R_d)\big)\left|\frac{dx}{dR_d}\right|, \tag{8}$$

where on the right-hand side the PDF of soil moisture is evaluated at moisture values corresponding to given respiration values. This is done by inverting Eq. (7) and expressing $x$ as a function of $R_d$,

$$x(R_d) = 1 - \sqrt{1 - \frac{R_d}{R_{d,max}}}. \tag{9}$$

We note that Eq. (7) is monotonic in the domain $0 \leq x \leq 1$, which allows defining unambiguously the inverse of $R_d(x)$. Had we used a non-monotonic $R_d(x)$ function (e.g., for applications of this approach to soils experiencing long saturation periods), the derived distribution approach would have required splitting the $x$ domain into two—one for each monotonic branch of $R_d(x)$. In turn, Eq. (9) allows calculating the slope of the $x(R_d)$ relation, which is also needed in Eq. (8),

$$\frac{dx}{dR_d} = \left(2R_{d,max}\sqrt{1 - \frac{R_d}{R_{d,max}}}\right)^{-1}. \tag{10}$$

The PDF of $R_d$ is thus obtained from Eq. (8)-(10) as

$$p_{R_d}(R_d) = C_1 \frac{e^{-\gamma\left(1 - \sqrt{1-r_d}\right)}\left(1 - \sqrt{1-r_d}\right)^{-1+\frac{\lambda}{\eta}}}{2\eta R_{d,max}\sqrt{1-r_d}}, \tag{11}$$

where the normalized respiration $r_d = \frac{R_d}{R_{d,max}}$ is introduced to simplify the notation. This PDF can now be used to calculate analytically the long-term mean of $R_d$, denoted by $\langle R_d \rangle$,

$$\langle R_d \rangle = \frac{R_{d,max}C_2}{\gamma^2}\left\{\Gamma\left[2 + \frac{\lambda}{\eta}, \gamma\right] - 2\gamma\Gamma\left[1 + \frac{\lambda}{\eta}, \gamma\right] - \frac{\lambda(\eta - 2\gamma\eta + \lambda)}{\eta^2}\Gamma\left[\frac{\lambda}{\eta}\right]\right\}, \tag{12}$$

where for convenience the parameter group $C_2 = \left(\Gamma\left[\frac{\lambda}{\eta}\right] - \Gamma\left[\frac{\lambda}{\eta}, \gamma\right]\right)^{-1}$ is defined. The standard deviation of $R_d$, denoted by $\sigma_{R_d}$, can not be obtained analytically, but it can be calculated through numerical integration of Eq. (11).

### 2.1.3. Heterotrophic respiration pulses at rewetting

Heterotrophic respiration pulses at rewetting are caused by mineralization of available C and microbial products at the end of the dry period, which in turn depend on how intense the rewetting event was. As a result of these processes, in a given soil, rewetting events depend on both soil moisture before the rewetting $x_d$, and the change in soil moisture $y$ (Birch, 1958; Lado-Monserrat et al., 2014). This relation can be captured by the empirical function (justified and parameterized in Section 2.2.1),

$$R_r = R_{r,max}\frac{y}{1 + \frac{x_d}{b}}\theta[(1 - x_d) - y], \tag{13}$$

where $R_{r,max}$ is the largest respiration pulse possible, which is achieved when an initially dry soil reaches saturation; i.e., $y=1$ and $x_d=0$. The parameter $b$ accounts for the effect of antecedent soil moisture conditions—for a given value of $x_d$, the respiration pulse increases with increasing $b$. The last term in Eq. (13) is a Heaviside function limiting the relation between $R_r$ and $y$ to conditions in which soil moisture at most fills the available pore space (as in Eq. (3), $\theta[\cdot]$ is equal to one only when $y < 1 - x_d$). If before the rain event soil moisture is at the plant wilting point ($x_d=0$) and the precipitation event is sufficient to reach $s_1$ (i.e., $y = x - x_d = 1$), the maximum respiration pulse is attained and $R_r=R_{r,max}$. Here, $R_r$ represents an amount of C respired when the rewetting event occurs, so its dimensions differ from those of the respiration rate during drying, $R_d$; these two quantities are combined in the total heterotrophic respiration rate in Section 2.1.5.

Because both $y$ and $x_d$ are random variables that follow the PDF of Eq. (6), also $R_r$ should be regarded as a random variable following its own PDF. Different from the PDF of $R_d$, which was obtained from the univariate PDF of soil moisture, the PDF of $R_r$ has to be derived from the joint PDF of $y$ and $x_d$. The derived distribution approach can still be used, but it requires the determinant of the Jacobian matrix of the transformation from $y$ and $x_d$ to $R_r$ (Kottegoda and Rosso, 1998). To proceed, it is first convenient to introduce an auxiliary variable $X=x_d$, which is used together with Eq. (13) to find the transformation from the original variables $y$ and $x_d$ to $R_r$ and $X$,

$$\begin{cases} X = x_d \\ R_r = R_{r,max}\frac{y}{1 + \frac{x_d}{b}} \end{cases} \Rightarrow \begin{cases} x_d = X \\ y = \frac{R_r}{R_{r,max}}\left(1 + \frac{X}{b}\right) \end{cases} \quad \text{for} \quad y < 1 - x_d, \tag{14}$$

where the inequality limits the soil-moisture increments as the Heaviside function in Eq. (13). Second, the system on the left of Eq. (14) is inverted to express the original variables as a function of the transformed variables (reported on the right of Eq. (14)), similar to the inversion done in Eq. (9). Third, we calculate the Jacobian matrix,

$$J = \begin{bmatrix} \frac{\partial x_d}{\partial X} & \frac{\partial x_d}{\partial R_r} \\ \frac{\partial y}{\partial X} & \frac{\partial y}{\partial R_r} \end{bmatrix} = \begin{bmatrix} 1 & 0 \\ \frac{R_r}{R_{r,max}} \frac{1}{b} & \frac{1}{R_{r,max}} \left(1 + \frac{X}{b}\right) \end{bmatrix},$$

(15)

and the determinant of the Jacobian,

$$|J| = \frac{1}{R_{r,max}} \left(1 + \frac{X}{b}\right).$$

(16)

Fourth, the joint PDF of the variables $X$ and $R_r$ is obtained using the derived distribution approach,

$$p_{X,R_r}(X, R_r) = p_{y,x_d}\big(y(X, R_r), x_d(X, R_r)\big)|J|,$$

(17)

where as in Section 2.1.2 all the terms on the right-hand side only depend on $X$ and $R_r$, and $p_{y,x_d}$ is given by Eq. (6). Finally, to obtain the (marginal) PDF of $R_r$, the joint PDF in Eq. (17) is integrated over all possible values of $X$,

$$p_{R_r}(R_r) = \int_0^1 p_{X,R_r}(X, R_r)\, dX =$$

$$= \frac{c'}{(b+r_r)R_{r,max}} \left\{ e^{-\gamma} \left[ \frac{\gamma(1-r_r)}{1+\frac{r_r}{b}} \right]^{\frac{\lambda}{\eta}} \frac{1+b}{1-r_r} + \right.$$

$$\left. e^{-\gamma r_r} \left( \frac{1}{1+\frac{r_r}{b}} \right)^{\frac{\lambda}{\eta}} \left[ \gamma(b + r_r) \left( \Gamma\left[\frac{\lambda}{\eta}\right] - \Gamma\left[\frac{\lambda}{\eta}, \gamma(1 - r_r)\right] \right) + \Gamma\left[1 + \frac{\lambda}{\eta}\right] - \Gamma\left[1 + \frac{\lambda}{\eta}, \gamma(1 - r_r)\right] \right] \right\}$$

(18)

where on the right hand side the normalized respiration pulse $r_r = \frac{R_r}{R_{r,max}}$ is introduced to simplify the notation, and as before $C_2 = \left( \Gamma\left[\frac{\lambda}{\eta}\right] - \Gamma\left[\frac{\lambda}{\eta}, \gamma\right] \right)^{-1}$. Due to the complexity of Eq. (18), the long-term mean and standard deviation of $R_r$, respectively

denoted by $\langle R_r \rangle$ and $\sigma_{R_r}$, need to be obtained via numerical integration.

### 2.1.4. Rewetting pulses only dependent on soil moisture change

It is useful to consider respiration pulses that only depend on the soil moisture increments; i.e., $b \gg x_d$. In this case, Eq. (13) reduces to $R_r = y R_{r,max}$ (i.e., $y = R_r/R_{r,max}$)—equivalent to having always a completely dry soil before rewetting. Thanks to the simplicity of the respiration pulse equation, $p_{R_r}(R_r)$ can be obtained as a derived distribution from the marginal PDF of

the soil moisture changes $y$,

$$p_y(y) = \int_0^1 p_y(y|x_d)dx_d = [1 + \gamma(1 - y)]e^{-y\gamma},$$

(19)

where $p_y(y|x_d)$ is from Eq. (3). The $p_{R_r}(R_r)$ is then obtained as,

$$p_{R_r}(R_r) = p_y\big(y(R_r)\big)\left|\frac{dy}{dR_r}\right| = \frac{1+\gamma\left(1-\frac{R_r}{R_{r,max}}\right)}{R_{r,max}} e^{-\frac{\gamma R_r}{R_{r,max}}}.$$

(20)

Thanks to the simplicity of Eq. (20), in this particular case the long-term mean and standard deviation of the respiration pulses are found analytically,

$$\langle R_r \rangle = \frac{R_{r,max}}{\gamma^2}(e^{-\gamma} + \gamma - 1),$$

(21)

$$\sigma_{R_r} = \frac{R_{r,max}}{\gamma^2}\sqrt{(\gamma - 2)\gamma - 1 + 2e^{-\gamma}(1 + \gamma + \gamma^2) - e^{-2\gamma}}.$$

(22)

Thus, when respiration pulses are simply proportional to the soil moisture change at rewetting, their mean only depends on the

maximum pulse size $R_{r,max}$ and the ratio of soil water storage capacity and mean precipitation depth (i.e., the parameter group $\gamma = \frac{nZ_r(s_1 - s_w)}{\alpha}$).

### 2.1.5. Combining heterotrophic respiration during soil drying and at rewetting

The total mean heterotrophic respiration rate is given by the sum of the mean respiration rate during soil drying $\langle R_d \rangle$ (Eq. (12); expressed in gC m$^{-2}$ d$^{-1}$) and the mean rate of respiration resulting from the sequence of rewetting pulses over the study period (denoted by $\langle R_r^* \rangle$ and also expressed in gC m$^{-2}$ d$^{-1}$). The $\langle R_r^* \rangle$ is calculated as the mean amount of respired carbon ($\langle R_r \rangle$ from Eq. (18), expressed in gC m$^{-2}$) divided by the mean rainfall inter-arrival time, $1/\lambda$ (expressed in days),

$$\langle R_r^* \rangle = \lambda \langle R_r \rangle. \tag{23}$$

The mean total heterotrophic respiration rate is then obtained as,

$$\langle R_t \rangle = \langle R_d \rangle + \langle R_r^* \rangle. \tag{24}$$

In what follows, the ratio of respiration pulse to total respiration (i.e., $\langle R_r^* \rangle / \langle R_t \rangle$) will also be considered, to evaluate the overall contribution of respiration pulses.

## 2.2. Data analysis

### 2.2.1. Laboratory incubation data for model calibration

The phenomenological respiration models in Eq. (7) and (13) require knowledge of three parameters: the heterotrophic respiration rate at the soil field capacity ($R_{d,max}$), the maximum respiration pulse size ($R_{r,max}$), and the sensitivity of the respiration pulse to the initial soil moisture ($b$). To estimate these three parameters, we selected datasets where both the soil moisture before rewetting and the soil moisture increments were manipulated (Fischer, 2009; Guo et al., 2014; Lado-Monserrat et al., 2014). All data reported in these three publications were used, except data from the litter-amended soils in Lado-Monserrat et al. (2014) (we chose to focus on 'natural' conditions) and data from small ($y<0.3$) rewetting events in Fischer (2009) (they exhibited small respiration peaks despite nearly stable soil moisture). The reported respiration amounts at rewetting were corrected to isolate the pulse size ($R_r$) from the respiration that would have occurred at constant soil moisture ($R_d$). This was done by calculating $R_{d,max}$ from control soil samples kept constantly wet (Guo et al., 2014) or from the post-pulse respiration rate before soil moisture started to decline in experiments where drying was allowed in all samples (Lado-Monserrat et al., 2014). In contrast, respiration pulses had already been isolated by Fischer (2009). The last step of the parameter estimation involved fitting Eq. (13) to the data using a nonlinear least square algorithm (*fminunc* function in Matlab, R2018b, The MathWorks, Inc.).

Because respiration amounts and rates in these laboratory incubations were expressed respectively in µg g$^{-1}$ and µg g$^{-1}$ d$^{-1}$ (or on a per unit soil organic C basis), units were converted to g m$^{-2}$ and g m$^{-2}$ d$^{-1}$ using bulk densities and sampling depths reported in the original publications (results are shown in Table 2).

### 2.2.2. Field data for model validation

In addition to estimating the values of the three parameters in Eq. (7) and (13), we validated the results from the whole stochastic model by comparting the predicted long-term mean heterotrophic respiration rates to observations along a rainfall manipulation gradient in a semi-arid steppe (Zhang et al., 2017b, 2019). Briefly, the precipitation gradient was established by excluding 30% and 60% of precipitation with rain shelters, and by increasing precipitation by 30% and 60% through irrigation. By design, only precipitation amounts (not timing) were altered, resulting in five mean rainfall depths $\alpha$ =2.6, 3.9, 5.1, 6.4, and 7.6 mm. Mean evapotranspiration rates, soil moisture, and heterotrophic respiration rates along the rainfall gradient were obtained from the published supplementary materials in Zhang et al. (2019) or from the Dryad dataset by Zhang et al. (2017a). Hydrologic parameters that were not provided were estimated as follows. The maximum evapotranspiration rate (assumed

equal the potential evapotranspiration) and the mean rainfall frequency were estimated from May-August CRU data at the rainfall manipulation site ($E_{max}$=4.3 mm d$^{-1}$ and $\lambda$=0.41 d$^{-1}$). The soil at the site has sandy loam texture (Bingwei Zhang, personal communication) and soil properties were obtained accordingly: $n$=0.42, $s_w$=0.11, $s_1$=0.52 (Table 2.1 in Rodriguez-Iturbe and Porporato, 2004). Finally, the rooting depth $Z_r$=0.2 m was estimated as the soil depth above which approximately 70% of belowground productivity occurs, based on data from Zhang et al. (2020).

Regarding the parameters of the rewetting respiration function (Eq. (13)), we assumed $R_{d,max}$=2 gC m$^{-2}$ d$^{-1}$ and $b$=0.1. These values are deemed reasonable for mineral soils based on Table 2, and accounting for a rooting depth about double the sampling depth of the incubation experiments (which doubles the $R_{d,max}$ values in Table 2). Without specific information on respiration pulse sizes, we let $R_{r,max}$ vary over a wide range. Additionally, we tested the simplified respiration model (Section 2.1.4), which does not require any assumption on $b$, against the same total heterotrophic respiration dataset.

## 3.    Results

### 3.1.    Dependence of heterotrophic respiration at rewetting on soil moisture

Laboratory incubation data were used to parameterize the functions linking heterotrophic respiration to soil moisture. As expected, the respiration pulses at rewetting depend on both rewetting intensity ($y$) and pre-wetting soil moisture ($x_d$), and this relation is well-characterized by Eq. (13) (Figure 2). In Figure 2, respiration pulses at rewetting are normalized by the amount of organic C in each soil to facilitate comparisons. However, after accounting for variations in organic C content, bulk density and soil layer depth, the values of $R_{r,max}$ and $R_{d,max}$ per unit ground area are higher in the organic soils than in mineral soils (Table 2), and so is the ratio between $R_{r,max}$ and $R_{d,max}$. The sensitivity parameter $b$ shows milder variation across soils than the other parameters, with an average value $b \approx 0.1$. Based on this data analysis, in the following theoretical exploration we set parameter values intermediate between the extremes reported in Table 2 (i.e., $R_{r,max}$=5 gC m$^{-2}$, $R_{d,max}$=1 gC m$^{-2}$ d$^{-1}$, and $b$=0.1). In addition, we explore how the contribution of respiration pulses varies between mineral vs. organic soils, using the average parameter values reported in Table 2.

### 3.2.    General model behaviour

Figure 3 shows two examples of the simulated trajectories of soil moisture and heterotrophic respiration, for contrasting climatic conditions (more frequent precipitation in the left panels than in the right panels). It is important to note that in this comparison across climatic conditions (and in the comparisons that follow), the maximum respiration $R_{r,max}$ and $R_{d,max}$ are fixed, while in reality they are likely proportional to soil organic C availability, which in turn is the result of a long-term and soil moisture-dependent balance between C inputs from vegetation and respiration (this limitation is discussed in Section 4.2). Respiration rates during dry periods follow soil moisture changes, declining as soil dries and returning to higher levels at rewetting (Figure 3b, f). In addition to this rewetting-induced restoration of high respiration rates, rewetting causes $CO_2$ emission pulses, represented by vertical bars. Under the wetter climate (Figure 3b), respiration pulses are more frequent than under the dry climate (Figure 3f) because of the higher precipitation frequency. However, most of the respiration pulses are small because soil moisture increments at rewetting are often limited by the available soil pore space and a relatively large fraction of precipitation is lost to runoff and deep percolation. In contrast, under dryer conditions, changes in soil moisture are large because on average soil moisture is low and the pore space is rarely filled up completely. As a result, the fewer respiration pulses can be larger under dry than under wet conditions.

The bottom panels in Figure 3 show the PDF of respiration for the same two climatic conditions analysed in the upper panels. While the PDF of $R_r$ is positively skewed regardless of climate (but with heavier tails under dry conditions, Figure 3c, g), the PDF of $R_d$ is strongly affected by climatic conditions—the probability of high values for $R_d$ is higher under wet conditions

(negatively skewed PDF) and lower under dry conditions (positively skewed PDF, Figure 3d, h). This pattern is caused by the prevalence of high soil moisture values in the wet climate scenario, which maintain relatively high $R_d$. Figure 3c, d, g, h also show that the theoretical PDF (Eq. (11) and (18)) match perfectly the distribution of the numerically simulated data. The shape of the theoretical PDF of $R_d$ in Figure 3h might seem incorrect, as it increases sharply at high respiration values. This increase is due to the flat derivative of the $R_d$-soil moisture relation (Eq. (7)), which causes an asymptote in the PDF at $R_d=R_{d,max}$ (Eq. (11)). However, the area under this spike is vanishingly small when climatic conditions are dry as in the example of Figure 3e-h, so that it is highly unlikely to have any respiration value around $R_{d,max}$.

### 3.3. Model test under field conditions

Field data were used to test if the hydrologic and soil respiration models could capture trends in the mean evapotranspiration and heterotrophic respiration along a precipitation gradient (Figure 4). The trend of the mean evapotranspiration rate with increasing mean rainfall depth was captured reasonably well (Figure 4a), considering that no formal calibration was conducted, and all parameters were estimated based on independent information. Similarly, the model correctly predicts the trend in soil moisture (not shown), but with an overestimation bias around 0.05-0.1 (in terms of normalized soil moisture $x$). This overestimation is expected, because soil moisture had been measured in the drier top 0.1 m of soil, while the model considers average soil moisture over a 0.2 m depth. Also the trend in total heterotrophic respiration is predicted correctly by the full model, which explains 77% of the variance in the respiration data (black curve in Figure 4b). Calibrating the two parameters of Eq. (13) and $R_{d,max}$ would allow a better fit, but since the goal here is to provide a qualitative model validation and not a quantitative performance assessment, we deem the model suitable for the following theoretical analyses.

We also tested the simpler version of the model, in which respiration pulses only depend on the soil moisture increment. Without the effect of pre-wetting soil moisture, this version predicts higher mean respiration than the full model (red lines in Figure 4b), and higher contribution of rewetting respiration to the total heterotrophic respiration (red lines in Figure 4c).

### 3.4. Dependence of heterotrophic respiration on rainfall statistical properties

Figure 5 shows the predicted effect of precipitation regimes on heterotrophic respiration during drying and at rewetting (Figure 5a, b), on the total heterotrophic respiration rate (Figure 5c), and on the fraction of respiration contributed by rewetting pulses (Figure 5d). As in Figure 3, $R_{r,max}$ and $R_{d,max}$ are fixed to focus on the role of climatic conditions, so the patterns shown in Figure 5 should be interpreted as changes of mean respiration rates along gradients of precipitation frequency ($\lambda$) and mean depth ($\alpha$) for given soil organic C stocks. Because in this minimal model the mean precipitation rate is given by $\langle P \rangle = \alpha\lambda$, precipitation can be increased by assuming more frequent rain events (i.e., increasing $\lambda$), larger events (i.e., increasing $\alpha$), or both. Any of these changes increase mean respiration during drying and at rewetting (Figure 5a, b). As $\langle R_d \rangle$ increases with precipitation more than $\langle R_r^* \rangle$, the relative contribution of respiration pulses to the total respiration rate, $\langle R_r^* \rangle/\langle R_t \rangle$, tends to decrease from drier to wetter conditions, especially when rain events become more frequent (as opposed to more intense) (Figure 5d). This pattern is caused by the relatively larger respiration pulses occurring when soils are dry and rewetting causes large soil moisture increments (compare examples in Figure 3b and 3f). Moreover, the relative change of $\langle R_r^* \rangle/\langle R_t \rangle$ is smaller than the change in $\langle R_d \rangle$ or $\langle R_r^* \rangle$ as precipitation regimes are varied.

Not only the mean respiration rates vary with hydro-climatic conditions, but also the variability of both respiration rates during drying and respiration pulses at rewetting (Figure 6). The standard deviation of $R_d$ exhibits maxima at intermediate $\alpha$ when $\lambda$ is fixed, and at intermediate $\lambda$ when $\alpha$ is fixed (Figure 6a). This pattern is due to a shift in the shape of the PDF of $R_d$ when moving from dry to wet conditions. Under dry conditions, the PDF of $R_d$ has relatively low variance and is negatively skewed (Figure 3d); as conditions become wetter the PDF flattens and the variance increases, and finally under wet conditions the PDF transitions again to a low-variance, but positively skewed PDF (Figure 3h). In contrast, the PDF of $R_r$ is always positively

skewed with variance decreasing with increasing rainfall frequency (Figure 6b; compare examples in Figure 3c and g). However, increasing $\alpha$ for fixed $\lambda$ is predicted to increase the variance of $R_r$. The coefficients of variation (CV) of $R_d$ and $R_r$ vary less than the corresponding standard deviations and tend to decrease as conditions move from dry to wet (Figure 6c, d). Specifically, the CV of $R_d$ decreases with both increasing $\lambda$ and increasing $\alpha$. In contrast, the CV of $R_r$ is nearly independent of $\alpha$, but decreases with increasing $\lambda$.

### 3.5. Effects of rainfall intensification and organic C availability on heterotrophic respiration

Results shown in Figures 5 and 6 are based on average respiration model parameters; here, we explore how changing organic C content from mineral to organic soils affects the contribution of rewetting pulses to total soil heterotrophic respiration. We also focus on changes in respiration patterns along gradients of rainfall intensification; i.e., decreasing precipitation frequency $\lambda$ while precipitation event depth $\alpha$ is increased and total precipitation is kept fixed (as along the white contour curves in Figures 5 and 6). Figure 7 shows that rainfall intensification decreases $\langle R_t \rangle$ (Figure 7a), but increases $\langle R_r^* \rangle / \langle R_t \rangle$ (Figure 7b), regardless of soil organic C availability (black vs. grey curves) and total precipitation (dashed vs. solid curves). However, for a given total precipitation, organic soils (grey curves) exhibit both higher $\langle R_t \rangle$ and higher $\langle R_r^* \rangle / \langle R_t \rangle$ than mineral soils (black curves), due to their higher $R_{r,max}$ (Table 2). As a result, in organic soils, the contribution of respiration pulses can be as high as 20% of the total heterotrophic respiration, whereas in mineral soils it tends to be lower than 10%. Moreover, in both soils, higher precipitation increases $\langle R_t \rangle$ while decreasing $\langle R_r^* \rangle / \langle R_t \rangle$ (compare solid *vs.* dashed curves).

### 4. Discussion

Heterotrophic respiration fluctuates at multiple temporal scales in response to hydro-climatic variability (Messori et al., 2019; Rubio and Detto, 2017)—from inter-annual variations due to climatic anomalies and extreme events (Reichstein et al., 2013), to seasonal variations partly linked to plant activity (Zhang et al., 2018), to short-term fluctuations induced by soil drying and rewetting (Daly et al., 2009). Here we focus on respiration fluctuations during drying-wetting cycles, and how they are affected by precipitation regimes. Differently from most other modelling approaches to describe these dynamics, we develop a probabilistic model with analytical solutions for the probability density function of respiration rate (discussed in Section 4.1). For the sake of analytical tractability, this model rests on important assumptions (Section 4.2), but despite its simplicity has the potential to assess the effect of precipitation variability (and its expected changes) on heterotrophic respiration (Section 4.3).

### 4.1. Comparison with previous stochastic approaches

Most biogeochemical models assume that heterotrophic respiration (and other processes) depend on a generic soil property $\varphi$ following an empirical function $f(\varphi)$ (Bauer et al., 2008; Moyano et al., 2013). As $\varphi$ changes through time (e.g., soil moisture and temperature), also the biogeochemical rate associated with $\varphi$ varies. Thus, the biogeochemical models use the function $f(\varphi)$ to convert measured time series of soil moisture and other environmental variables into biogeochemical rates. The different approach we follow here consists in linking a known probability density of $\varphi$, to the probability density of the function $f(\varphi)$ to capture the propagation of the statistical properties of $\varphi$ to $f(\varphi)$. This can be done by the derived distribution approach, as in Eq. (8). This approach has been used to investigate gaseous nitrogen emissions in response to soil moisture fluctuations (Ridolfi et al., 2003), but the only example studying soil heterotrophic respiration rate we are aware of focused on respiration responses to temperature fluctuations (Sierra et al., 2011). These approaches provide simple and mathematically elegant solutions, but have so far been limited to the effect of a single driver of the biogeochemical flux of interest. The responses of heterotrophic respiration to changes in soil moisture are more complex because rewetting pulses depend on both soil moisture

increment and pre-wetting soil moisture (Figure 2), requiring the solution of a bivariate stochastic process. Thus, our approach—by accounting for both these effects—is more general and applicable along gradients where the statistical properties describing the precipitation regime vary significantly (Figure 4).

A previous stochastic approach focused on the $CO_2$ concentration in the pore space instead of respiration rates (Daly et al., 2008). Observations show that $CO_2$ concentration increases rapidly after rainfall, and then decreases following a negative exponential function. This dynamic can be described as a stochastic process where $CO_2$ concentration is the random variable and precipitation represents the stochastic forcing (Daly et al., 2008). With this approach, the long-term mean $CO_2$ concentration was found to depend on the average rainfall rate ($\lambda\alpha$), while the standard deviation of $CO_2$ concentration depends on $\lambda\alpha^2$. This indicates that rainfall intensity (in terms of mean event depth $\alpha$) plays a more important role than rainfall frequency in driving the variability of soil $CO_2$ concentration. Soil respiration was shown to be approximately proportional to $CO_2$ concentration in the pore space over a broad range of concentrations (Daly et al., 2008), so that respiration statistics are also expected to scale with rainfall statistics in the same way as soil $CO_2$ concentrations. This result is consistent with our finding that all components of heterotrophic respiration increase with both $\lambda$ and $\alpha$ (Figure 5).

Numerical process-based models have also been driven by randomly-generated rainfall time series (e.g., Tang et al., 2019). These models do not allow finding analytical solutions for the respiration statistical properties, but offer insights on the individual processes affecting these properties. For scenarios of constant total rainfall and variable rain event frequency, Tang et al. (2019) found that rainfall intensification increased heterotrophic respiration in a semi-arid grassland, even though in their simulations, also soil organic C stocks slightly increased due to higher plant productivity. This result differs from our finding that total heterotrophic respiration decreases with rainfall intensification (moving right to left along the curves in Figure 7a), and was likely caused by how plant productivity and its feedback to soil organic C were modelled in their study.

### 4.2. Methodological limitations

Three model assumptions can alter the interpretation of our results: i) that heterotrophic respiration pulses can be regarded as instantaneous; ii) that the two parameters $R_{d,max}$ and $R_{r,max}$ are independent of climatic and vegetation conditions; and iii) that hydro-climatic conditions are statistically stationary.

Respiration pulses are modelled as instantaneous events of $CO_2$ emission with a given size (Section 2.1.3). While mathematically convenient, rewetting respiration pulses are known to last for a few days after the rewetting has ended. Indeed, when analysing laboratory incubation data, the pulse size is generally calculated by integrating through time the respiration rates above the rate occurring at stable soil moisture. The integration window ranges between two and three days (e.g., Fischer, 2009). This simplified approach to separate the actual rewetting pulse from the respiration rate at stable soil moisture requires some caution when rainfall events are frequent. In that case, pulses would overlap rather than being distinct. Moreover, with frequent rainfall, respiration could be inhibited due to water logging (Moyano et al., 2013; Rubio and Detto, 2017), and no respiration pulse might occur. Thus, to avoid these issues, our equations should not be used in wet environments with $\lambda >0.3$ $d^{-1}$.

We calculated the statistical properties of the heterotrophic respiration rate, but did not consider the dynamics of the soil organic matter and plants that supply resources for microbial growth and respiration. Widely different precipitation amounts and distributions such as those depicted in Figures 5 and 6 are associated with different plant communities, whose productivity increases along gradients of precipitation (Huxman et al., 2004; Luyssaert et al., 2007), providing litter and root exudates whose C is eventually stabilized into soil organic matter. Indeed, soil organic C stocks increase with increasing mean annual precipitation (Guo et al., 2006). Hence, soil organic matter probably varies along the axes of Figures 5 and 6, which are instead interpreted here as purely climatic gradients. Such variations in organic matter content would affect the maximum respiration rate and pulse size, $R_{d,max}$ and $R_{r,max}$ (e.g., compare mineral and organic soils in Table 2). Because the mean respiration rates scale with the maximum rates (as apparent analytically from Eq. (21)), it is reasonable to expect that higher organic matter

content along precipitation gradients increases the sensitivity of respiration to changes in precipitation compared to predictions in Figure 5. Indeed, even when keeping precipitation constant while varying the frequency and depth of precipitation events, the variations in total heterotrophic respiration are larger in organic soils than in mineral soils (Figure 7a).

Moreover, soil C substrates might be depleted through multiple drying and rewetting events—a behaviour we do not consider in the proposed statistically stationary model. While some experiments show sustained rewetting pulses (Miller et al., 2005; Xiang et al., 2008), others show reduced total heterotrophic respiration with increasing frequency of drying and rewetting, possibly due to substrate depletion (Shi and Marschner, 2014). To capture these dynamics, a more complex model describing the changes in substrate and microbial compartments would be needed (e.g., Brangarí et al., 2018; Lawrence et al., 2009; Tang et al., 2019) at the cost of losing the analytical tractability.

Our focus in this contribution is on heterotrophic respiration, but the data we used to parameterize the model are from laboratory studies without plants. Therefore, our heterotrophic respiration estimates neglect contributions from fresh C inputs from roots to the rhizosphere (Finzi et al., 2015; Kuzyakov and Gavrichkova, 2010). However, the timing of rhizodeposition depends on plant activity, which in turn depends on previous environmental conditions—differently from soil microbes that respond to soil moisture changes rapidly, plant responses integrate previous conditions thereby partly decoupling root activity from current soil moisture. It is thus non-trivial to include rhizosphere processes in the current framework.

In addition to these limitations, our results should also be interpreted with caution when rainfall seasonality is important, because the assumption of stochastic stationarity (Section 2.1.1) may not be met, requiring the derivation of a different probability density function of soil moisture (e.g., Vico et al., 2017). Nevertheless, our results will still hold for parts of the year when the rainfall regime is relatively stable.

### 4.3. How are the statistical properties of heterotrophic respiration varying with changing precipitation regimes?

The axes of Figures 5 and 6 can be interpreted in terms of changes in precipitation patterns caused by ongoing climatic changes. If rainfall in a semi-arid or mesic environment increases (due to either more frequent or larger events), heterotrophic respiration also increases (Yan et al., 2014; Zhang et al., 2019)—this is not surprising as soils become on average wetter, removing water limitation and promoting microbial activity. These observations are consistent with our findings that the mean respiration pulse at rewetting and respiration during drying increase with increasing $\alpha$, $\lambda$, or their product—i.e., total precipitation. However, the variability in respiration does not always change monotonically with increasing rainfall. Figure 6b shows that the standard deviation of the respiration pulses increases with more intense (higher $\alpha$) and less frequent (lower $\lambda$) rainfall. In contrast, the standard deviation of the respiration rate during drying, $R_d$, peaks at intermediate $\alpha$ and $\lambda$, and declines thereafter because the respiration response is flat and thus have higher variance at intermediate wetness (Eq. (7); Figure 6a). Therefore, higher precipitation as driven by increasing $\alpha$ or $\lambda$ is expected to increase the respiration pulses (Figure 5b) and their variability (Figure 6b), while decreasing their contribution to the total heterotrophic respiration (Figure 5d).

It is perhaps more interesting to understand respiration responses to changes in rainfall patterns for given total rainfall amounts. When $\alpha$ and $\lambda$ are changed simultaneously while keeping their product fixed (moving along the white curves in Figures 5-6; or along the x-axis in Figure 7), the mean respiration pulse at rewetting and the standard deviations of both respiration components increase with more intermittent and intense rainfall events. In experimental rainfall manipulations that mimic the predicted climatic changes, increased variability in soil moisture associated with more intense but less frequent precipitation events decreases total soil respiration (Harper et al., 2005). This observation is consistent with our result that the mean total heterotrophic respiration decreases with rainfall intensification while maintaining a given mean precipitation rate (i.e., moving right to left along the curves in Figure 7a). Our result is explained by the higher runoff and deep percolation losses predicted by the soil hydrologic model when precipitation events are large but rare (Rodriguez-Iturbe and Porporato, 2004). These water losses cause soil moisture to be on average lower as the precipitation regime becomes more intermittent—a pattern also confirmed empirically in rainfall manipulation experiments (Harper et al., 2005). Our approach neglects the lower plant C

inputs and contributions to total soil respiration under a more intermittent precipitation regime (Harper et al., 2005), which further reduces the total (combined autotrophic and heterotrophic) soil respiration rate.

We also found that the contribution of rewetting pulses to the total heterotrophic respiration increases when rainfall becomes more intermittent and rainfall events larger (i.e., moving right to left along the curves in Figures 7b). This result is consistent with observations in a temperate steppe (Yan et al., 2014). The rewetting pulse contribution is also larger in organic soils compared to mineral soils (gray vs. black curves in Figure 7)—this effect is expected, because more C can be mobilized by drying and rewetting cycles in C-rich soils (Canarini et al., 2017). We can thus surmise that climatic changes causing longer
dry period and more intense rainfall events (IPCC, 2012) will increase the role of pulse responses, including not only respiration, but also nitrogen mineralization pulses that could release nitrogen at a time when plant uptake is low. In turn, this can cause a de-coupling of nitrogen supply and demand, with possible negative consequences for ecosystem productivity (Augustine and McNaughton, 2004; Dijkstra et al., 2012).

Our findings are based on time-invariant relations between heterotrophic respiration and soil moisture, but temperature and
445 other environmental conditions also affect microbial activity—in part directly and in part indirectly via rhizodeposition— raising the question of how our results could be impacted by other respiration controlling factors. As a first approximation, temperature could be assumed to alter directly both respiration rates during drying and respiration pulses in similar way. This implies that our results would hold even under fluctuating temperatures, at least during the growing season, when temperature variations are limited and precipitation can be described by a simple marked Poisson process (Section 2.1.1). However, a
450 different modelling approach would be needed to quantify the mean heterotrophic respiration rate during seasons with frequent rainfall events, when respiration pulses are likely to be less important and anaerobic conditions (here neglected) could play a role. As the time scale expands from the growing season to the whole year, also seasonal fluctuations in plant activity that delay the supply of C substrates to microbes will play a role (Finzi et al., 2015; Kuzyakov and Gavrichkova, 2010), leading to a hierarchy of responses at multiple time scales—a more complex problem than the one addressed in this contribution.

**5. Conclusions**

Heterotrophic respiration depends nonlinearly on soil moisture—not only does it follow soil moisture during a dry period, but it also responds rapidly to rewetting. These rewetting responses occur in the form of pulses of $CO_2$ whose size increases with increasing soil moisture increment and decreasing pre-wetting soil moisture. We used this relation between respiration pulses and soil moisture to characterize analytically the statistical properties of respiration rates as a function of the statistical
properties of the rainfall events that drive soil moisture changes. Consistent with empirical evidence, our model predicts that dryer climatic conditions (either lower rainfall depths or longer dry periods between two rain events) lower total heterotrophic respiration. More interestingly, we showed that the contribution of rewetting pulses to the total heterotrophic respiration increases in dryer climates, but also when the precipitation regimes shift towards more intermittent and intense events (even at constant total average rainfall). Therefore, our results suggest that the expected intensification of precipitation will increase
the role of rewetting respiration pulses in the ecosystem C budgets.

**Data availability**

All data used in this study are published and available in the original publications and linked supplementary materials (see Table 2 for references on the laboratory data and Section 2.2.2 for references on the field data).

**Author contributions**

SM and GV conceptualized the study; SM, GV, and AP developed the theory; TF provided and discussed data; AC analysed data and prepared Fig. 2; SM prepared the other figures and drafted the manuscript; all authors discussed the study ideas, and read and commented the manuscript.

**Competing interests**

The authors declare that they have no conflict of interest.

**Acknowledgements**

This work was supported by the Swedish Research Council Vetenskapsrådet (grant 2016-04146) and Formas (grants 2018-00968 and 2018-00425). GV was partially supported by the Swedish Research Council Vetenskapsrådet (grant 2016-04910). We thank Antonio L. Lidón and Bingwei Zhang for sharing data and assisting in their interpretation, and Thomas Wutzler and two anonymous reviewers for their constructive comments.

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

**Table 1: Symbol definitions and units. Symbol $p_z$ represents the probability density function (PDF) of the stochastic variable $z$ indicated in the subscript.**

| Symbol | Definition | Units |
|---|---|---|
| $b$ | Parameter in the rewetting respiration equation | - |
| $C_1$ | Normalization constant in the soil moisture PDF | - |
| $C_2$ | Parameter group, $C_2 = \left( \Gamma\left[\frac{\lambda}{\eta}\right] - \Gamma\left[\frac{\lambda}{\eta}, \gamma\right] \right)^{-1}$ | - |
| $E$ | Evapotranspiration rate | m d$^{-1}$ |
| $E_{max}$ | Evapotranspiration rate at the soil field capacity | m d$^{-1}$ |
| $h$ | Precipitation event depth | m |
| $L$ | Rate of water loss via deep percolation and surface runoff | m d$^{-1}$ |
| $J$ | Jacobian matrix | |
| $n$ | Soil porosity | - |
| $p_{R_d}(R_d)$ | PDF of the soil respiration rate during dry-down periods | gC$^{-1}$ m$^2$ d |
| $p_{R_r}(R_r)$ | PDF of the respired carbon at rewetting | gC$^{-1}$ m$^2$ |
| $p_x(x)$ | PDF of normalized soil moisture ($x$) | - |
| $p_{x_d}(x_d)$ | PDF of normalized soil moisture at the end of the dry period ($x_d$) | - |
| $p_{X,R_r}(X, R_r)$ | Joint PDF of the auxiliary variable $X$ and of the respired carbon at rewetting ($R_r$) | gC$^{-1}$ m$^2$ d |
| $p_y(y)$ | Marginal PDF of soil moisture increase due to precipitation ($y$) | - |
| $p_y(y\|x_d)$ | PDF of soil moisture increase due to precipitation ($y$) conditional on soil moisture at the end of the previous dry period ($x_d$) | - |
| $p_{y,x_d}(y, x_d)$ | Joint PDF of soil moisture at the end of a dry period ($x_d$) and soil moisture increase due to precipitation ($y$) | - |
| $P$ | Precipitation rate | m d$^{-1}$ |
| $r_d$ | Normalized respiration rate during drying, $r_d = R_d/R_{d,max}$ | - |
| $r_r$ | Normalized respired carbon at rewetting, $r_r = R_r/R_{r,max}$ | - |
| $R_d$ | Respiration rate during dry-down periods | gC m$^{-2}$ d$^{-1}$ |
| $R_{d,max}$ | Maximum respiration rate at the soil field capacity | gC m$^{-2}$ d$^{-1}$ |
| $R_r$ | Respired carbon at rewetting | gC m$^{-2}$ |
| $R_{r,max}$ | Maximum respired carbon at rewetting (for $y$=1, $x_d$=0) | gC m$^{-2}$ |
| $\langle R_r^* \rangle$ | Mean rate of respiration from rewetting pulses | gC m$^{-2}$ d$^{-1}$ |
| $\langle R_t \rangle$ | Mean total respiration rate (sum of $\langle R_d \rangle$ and $\langle R_r^* \rangle$) | gC m$^{-2}$ d$^{-1}$ |
| $s$ | Relative volumetric soil moisture (i.e., saturation) | - |
| $s_w$, $s_1$ | Soil moisture at the wilting point and at field capacity, respectively | - |
| $t$ | Time | d |
| $x$ | Normalized soil moisture, $x = \frac{s - s_w}{s_1 - s_w}$ | - |
| $x_d$ | Normalized soil moisture at the end of a dry period | - |
| $X$ | Auxiliary variable, $X = x_d$ | - |
| $y$ | Change in normalized soil moisture at rewetting | - |
| $Z_r$ | Soil rooting depth | m |
| $\alpha$ | Mean precipitation event depth | m |

| $\gamma$ | Parameter group, $\gamma = \frac{n Z_r (s_1 - s_w)}{\alpha}$ | - |
|---|---|---|
| $\Gamma[\cdot]$ | Gamma function, $\Gamma[z] = \int_0^\infty u^{z-1} e^{-u} du$ | - |
| $\Gamma[\cdot,\cdot]$ | Incomplete gamma function, $\Gamma[a,z] = \int_z^\infty u^{a-1} e^{-u} du$ | - |
| $\delta[\cdot]$ | Dirac delta function | $[\text{argument}]^{-1}$ |
| $\eta$ | Parameter group, $\eta = \frac{E_{max}}{n Z_r (s_1 - s_w)}$ | - |
| $\lambda$ | Mean frequency of precipitation events | $d^{-1}$ |
| $\sigma_{R_d}$ | Standard deviation of the respiration rate during drying | $gC\ m^{-2}\ d^{-1}$ |
| $\sigma_{R_r}$ | Standard deviation of the respiration pulse at rewetting | $gC\ m^{-2}$ |
| $\theta[\cdot]$ | Heaviside step function | - |
| $\langle \cdot \rangle$ | Long term average | $[\text{argument}]$ |

**Table 2: Characteristics of the selected mineral and organic soil samples; estimates of the respiration model parameters in Eq. (13) ($R_{d,max}$: maximum respiration rate at the soil field capacity, $R_{r,max}$: maximum respired carbon at rewetting) and coefficients of determination ($R^2$) for the least square fit of the data (see also Figure 2).**

|  | Soil | Organic C (g/kg) | Bulk density (g/cm$^3$) | $R_{d,max}$ (gC m$^{-2}$ d$^{-1}$) | $R_{r,max}$ (gC m$^{-2}$) | $b$ (-) | $R^2$ | Source |
|---|---|---|---|---|---|---|---|---|
| Mineral soils | Chelva sandy loam | 10.9 | 1.44 | 0.79 | 0.89 | 0.17 | 0.74 | (Lado-Monserrat et al., 2014) |
|  | Tuéjar clay loam | 26.6 | 1.19 | 0.13 | 0.25 | 0.14 | 0.92 | (Lado-Monserrat et al., 2014) |
|  | Brookston clay loam | 28.6 | 1.24 | 1.49 | 8.79 | 0.04 | 0.76 | (Guo et al., 2014) |
|  | Average of mineral soils | 22.0 | 1.29 | 0.80 | 3.31 | 0.12 |  |  |
| Organic soils | Neuglobsow sand | 440 | 0.14 | 1.05 | 13.95 | 0.10 | 0.87 | (Fischer, 2009) |
|  | Taura silty sand | 390 | 0.15 | 1.26 | 11.23 | 0.10 | 0.86 | (Fischer, 2009) |
|  | Rösa sand | 340 | 0.18 | 1.53 | 18.61 | 0.12 | 0.83 | (Fischer, 2009) |
|  | Average of organic soils | 390 | 0.16 | 1.28 | 14.60 | 0.11 |  |  |

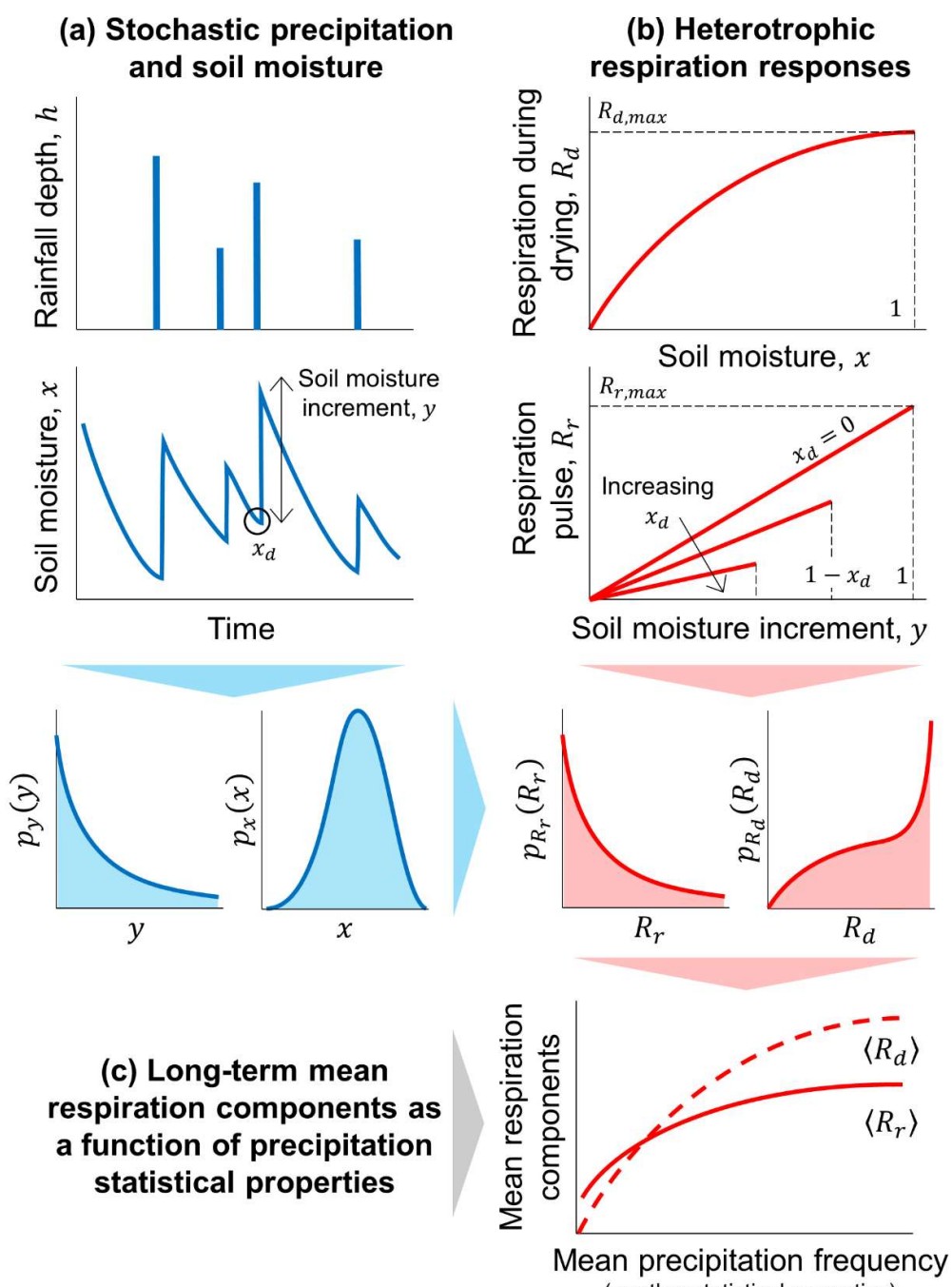

**Figure 1: Schematic illustration of the theoretical framework developed to describe how the components of heterotrophic respiration change as a function of rainfall statistical properties. (a) Rainfall is treated as a stochastic process driving random fluctuations in soil moisture, which are captured by the probability density functions (PDF, indicated by $p$ with a subscript for the variable of interest) of soil moisture ($x$) and soil moisture increments ($y$). (b) Respiration rate during drying ($R_d$) and respiration pulses at rewetting ($R_r$) respectively depend on soil moisture, and on both soil moisture increments and soil moisture at the end of the dry period ($x_d$); based on the PDF of $x$, $y$, and $x_d$, the PDF of the two respiration components are obtained. (c) Using these PDF of respiration, long term mean respiration rate during drying ($<R_d>$) and respiration pulses ($<R_r>$) are calculated and their relations with the statistical properties of precipitation are analysed.**

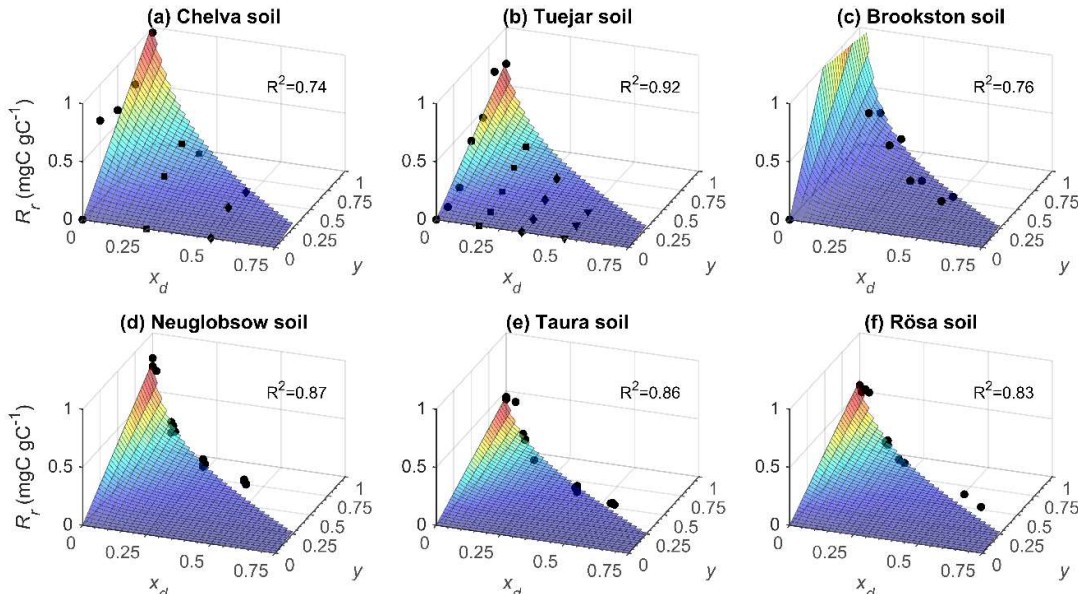

**Figure 2: Relations between respiration pulse size ($R_r$, normalized by soil organic C content) and pre-wetting soil moisture ($x_d$) and soil moisture increment at rewetting ($y$), for six soils; top row: mineral soils; bottom row: organic soils (note different vertical scales between rows). Symbols represent measured respiration pulses and surfaces are fitted $R_r$ functions from Eq. (13) (soil characteristics, fitting parameters, and data sources are reported in Table 2).**

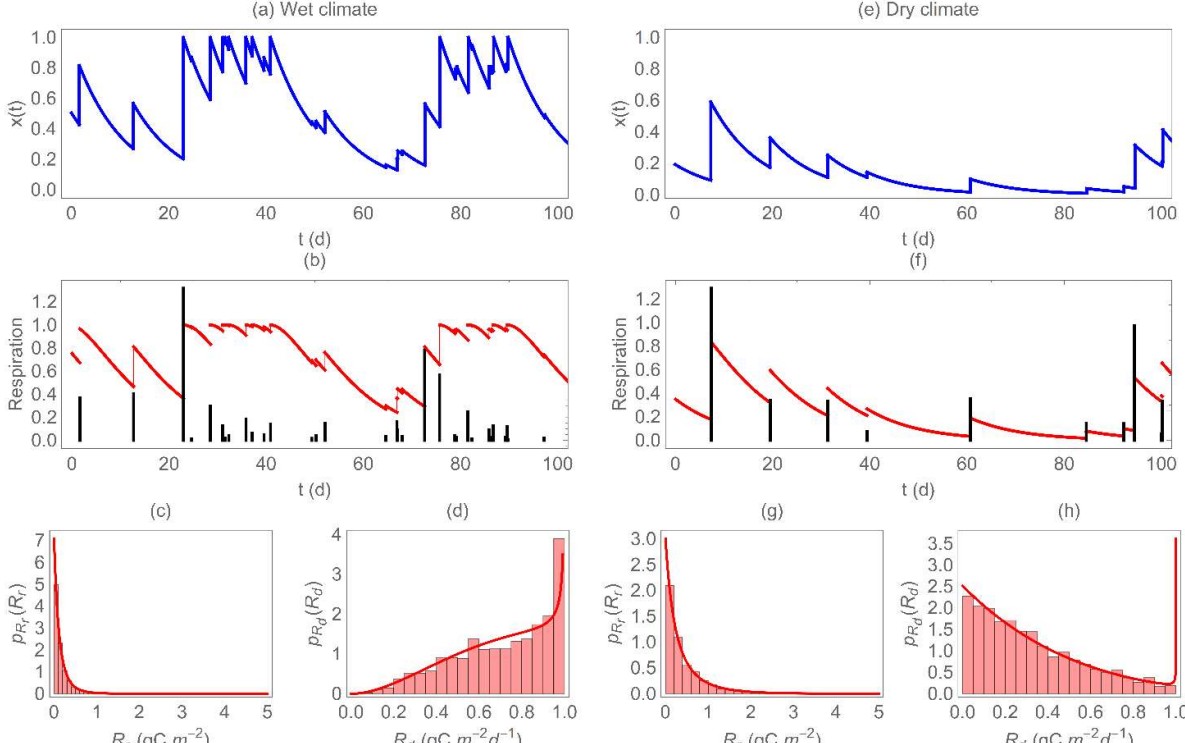

**Figure 3: Example of the dynamics of soil moisture and respiration for a wet (a-d) and a dry climate (e-h). Top panels (a, e) show the simulated trajectories of normalized soil moisture $x$; the middle panels (b, f) show the trajectories of**

675 **respiration during dry periods (red solid curves, $R_d$) and the respiration pulse at rewetting (black vertical bars, $R_r$, on the same scale despite different units); the bottom panels (c, d, g, h) show the probability density functions of $R_r$ and $R_d$ ($p_{R_r}(R_r)$ and $p_{R_d}(R_d)$, respectively) overlapped to the histograms of the simulated data. In this figure, $R_{r,max}$=5 gC m$^{-2}$, $R_{d,max}$=1 gC m$^{-2}$ d$^{-1}$, $b$=0.1, $\gamma$ =5, $\eta$ =0.1 d$^{-1}$, and $\lambda$ =0.3 and 0.1 d$^{-1}$ (panels a-d and e-h, respectively).**

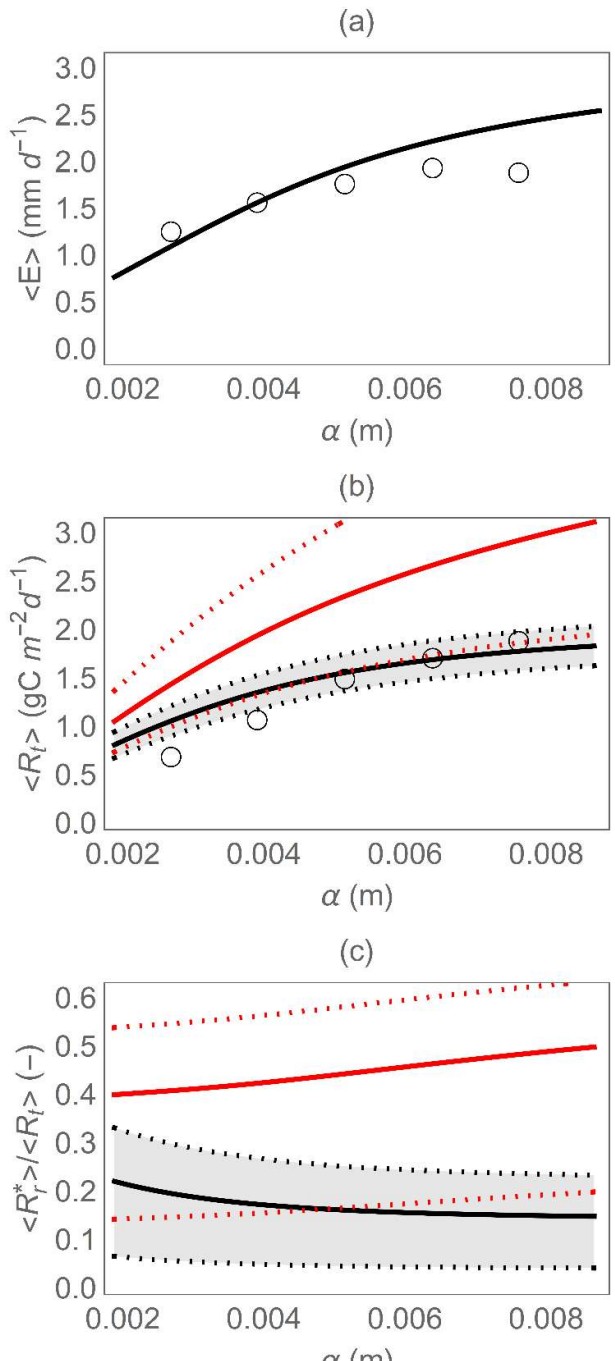

**Figure 4: Comparison of model results (curves) and observations (open circles) along an experimental rainfall gradient where the mean precipitation depth ($\alpha$) was manipulated: (a) mean evapotranspiration rate $\langle E \rangle$, (b) mean total heterotrophic respiration rate $\langle R_t \rangle$, and (c) fraction of the total heterotrophic respiration rate due to rewetting pulses $\langle R_r^* \rangle / \langle R_t \rangle$. In panels (b) and (c), the dotted curves and shaded area indicate the variation caused by changes in $R_{r,max}$**

**between 5 and 35 g m$^{-2}$ around the central value (solid curves) of 25 g m$^{-2}$; the red curves indicate results using the simplified rewetting respiration model (Eq. (21)). Parameter values are described in Section 2.2.2.**

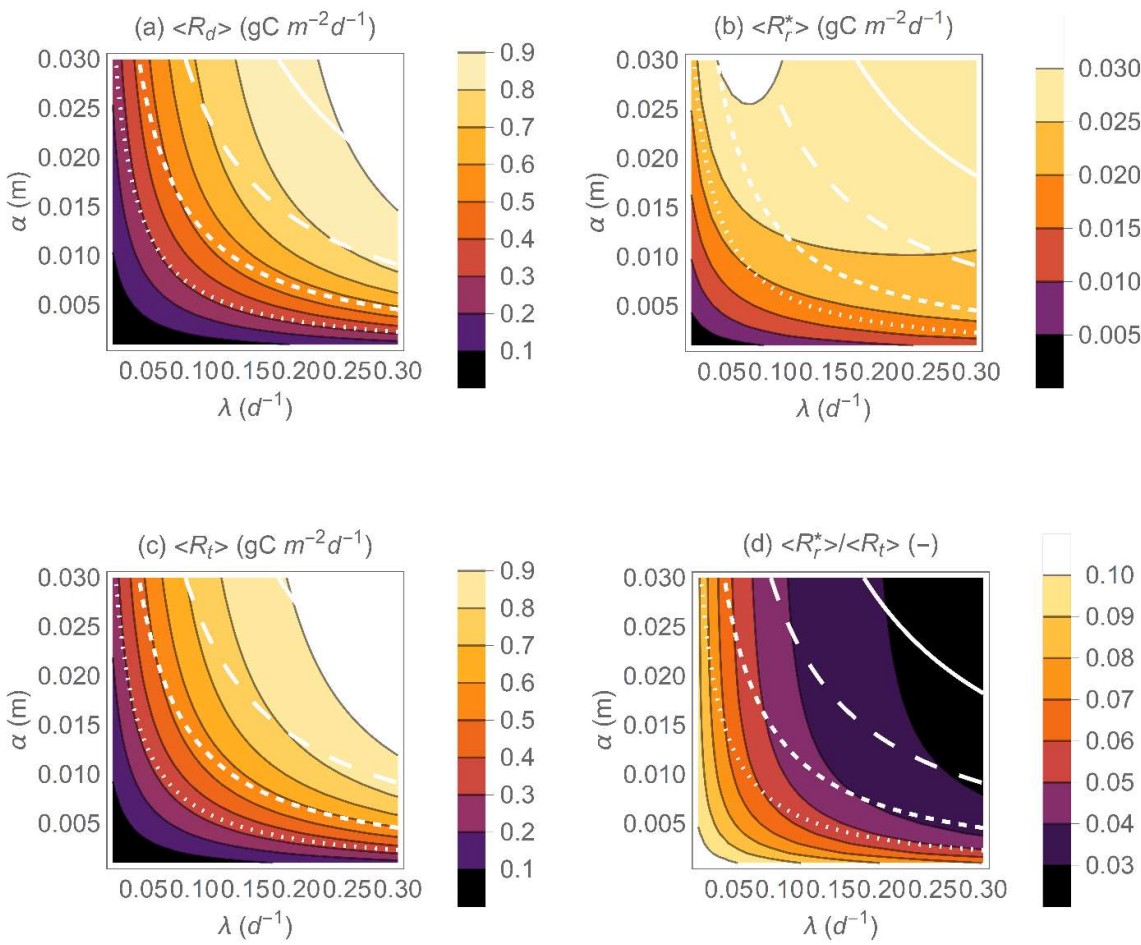

**Figure 5: Effect of precipitation statistical properties (mean event frequency $\lambda$ and depth $\alpha$) on the mean heterotrophic**
**respiration rates during dry periods $\langle R_d \rangle$ (a), and at rewetting $\langle R_r^* \rangle$ (b), the mean total respiration rate $\langle R_t \rangle$ (c), and on**
**the fraction of the total heterotrophic respiration rate due to rewetting pulses $\langle R_r^* \rangle / \langle R_t \rangle$ (d). The white contour curves**
**indicate combinations of $\lambda$ and $\alpha$ that generate different annual precipitation rates ($\langle P \rangle = \alpha\lambda$ =0.25, 0.5, 1, and 2 m y$^{-1}$ from dotted to solid lines). Other parameter values: $R_{r,max}$=5 gC m$^{-2}$, $R_{d,max}$=1 gC m$^{-2}$ d$^{-1}$, $b$=0.1, $Z_r$=0.3 m, $n$=0.5, $s_w$=0.2,**
**$s_1$=0.7, $E_{max}$=0.0037 m d$^{-1}$.**

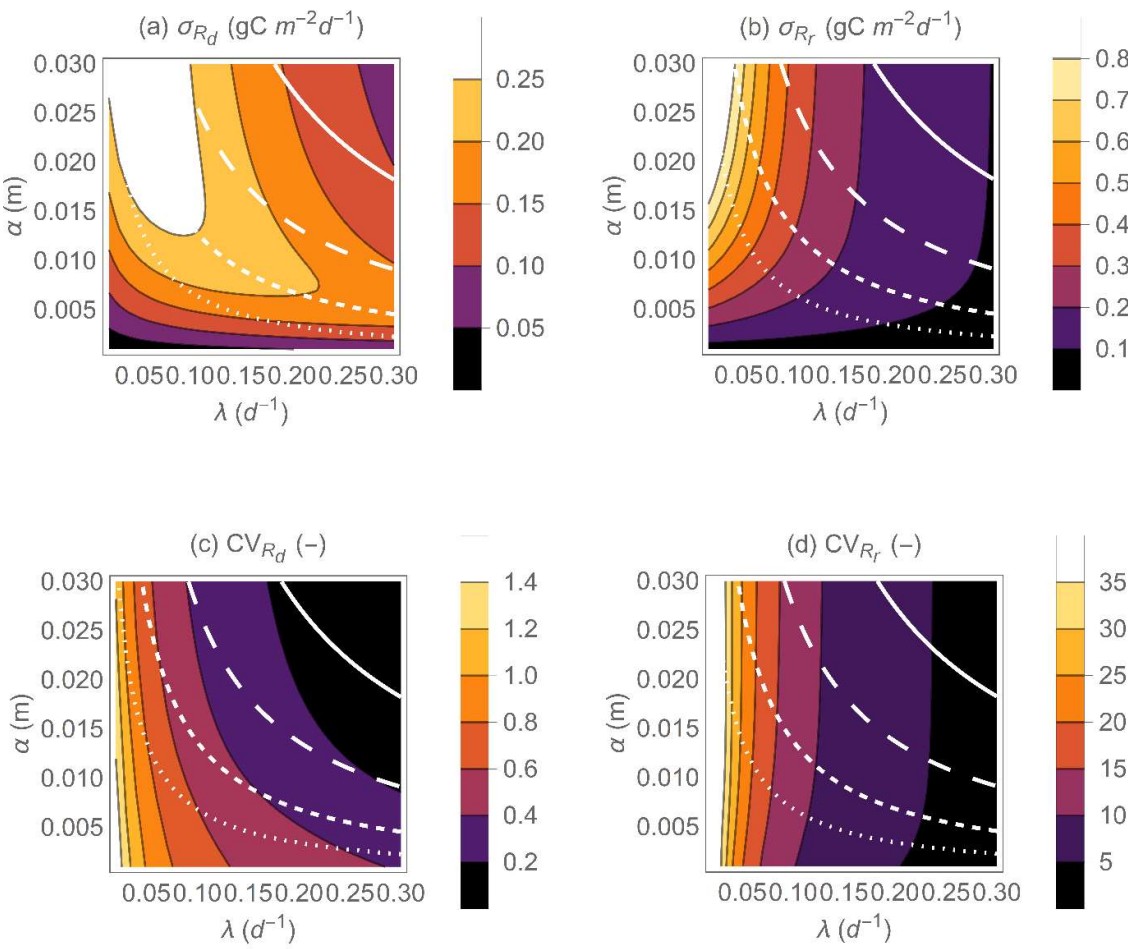

**Figure 6: Effect of precipitation statistical properties (mean event frequency $\lambda$ and depth $\alpha$) on the standard deviations of heterotrophic respiration rates during dry periods $\sigma_{R_d}$ (a) and respiration pulses at rewetting $\sigma_{R_r}$ (b), and on the coefficients of variations of respiration rates during dry periods $CV_{R_d}$ (c) and respiration pulses at rewetting $CV_{R_r}$ (d). The white contour curves indicate combinations of $\lambda$ and $\alpha$ that generate different annual precipitation rates ($\langle P \rangle = $**

**$\alpha\lambda$ =0.25, 0.5, 1, and 2 m y$^{-1}$ from dotted to solid lines). Other parameter values are as in Figure 5.**

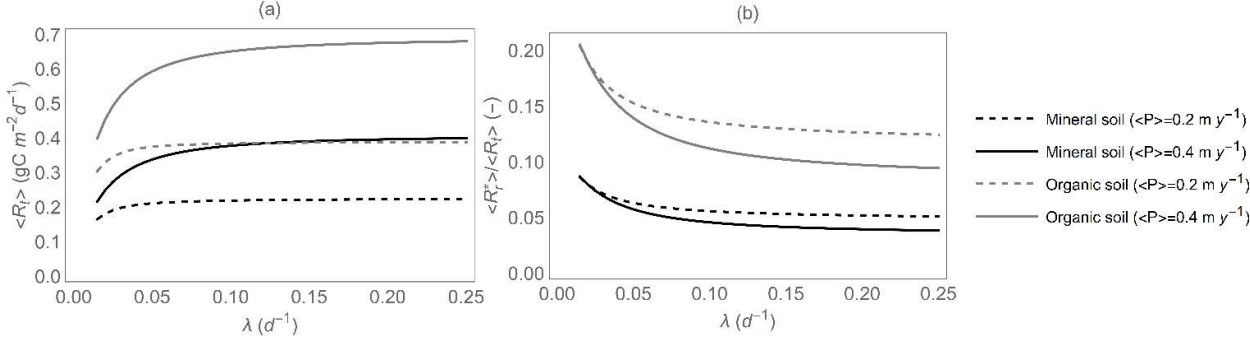

**Figure 7: (a)** Total heterotrophic respiration $\langle R_t \rangle$, and **(b)** fraction of the total heterotrophic respiration rate due to rewetting pulses, $\langle R_r^* \rangle / \langle R_t \rangle$, as a function of mean precipitation event frequency $\lambda$, for given total precipitation (i.e., $\alpha = \lambda / \langle P \rangle$), and for both mineral and organic soils. The respiration model parameters are reported in Table 2 and other parameter values are as in Figure 5.