# Peer review of "Rainfall intensification increases the contribution of rewetting pulses to soil heterotrophic respiration"

_Biogeosciences, 2020_

## Referee Comment (RC1) · Anonymous Referee #1 · 14 Apr 2020

I reviewed the manuscript n. bg-2020-95 entitled "Rainfall intensification increases the contribution of rewetting pulses to soil respiration". In this manuscript authors assess the validity of a stochastic model for soil heterotrophic respiration to explain the effects of precipitation variability on soil CO2 fluxes. The manuscript is very well written and the rationale for the need of this mode is well formulated. The theoretical considerations are sound, and the equations look correct. Authors also address the limits of their model in the discussion section (which is important when the readers will try to link to the model to their own experiment or for general extrapolation) and tend to not emphasize too much the findings of their model given the discussed limitation. Nevertheless the study is important because highlight an important aspect, which is the growing importance of rewetting events depending on precipitation distribution. This is an aspect

that is often overlooked, especially in manuscript dealing with drought, and highlight the importance of context dependent effects. I have no reserves in recommending this paper for publication, with only minor revisions and suggestions from my side. This has happened to me rarely and therefore I must compliment the authors for the excellent work.

GENERAL COMMENTS: My general comments are mainly small recommendation and suggestions:

1) I would recommend the authors to be more careful when they refer to soil respiration and to hererotrophic respiration. Indeed, in all their theoretical background, consideration is given only to microbial respired carbon and therefore neglecting the contribution of plant roots to drying and rewetting events. While this is fine for the paper, I would urge the authors to refer to heterotrophic soil respiration rather than to soil respiration mostly throughout the paper. 2) I understand that for simplicity authors calculate average parameters between soil with high OC and low OC (L. 233-235) for further theoretical considerations. However wouldn't be also very interesting to get parameter for both (high and low OC) and understand whether we can draw similar conclusions in both cases? Also this would make it more easy to then look at the larger picture, as you in part address (L. 356-358), that systems that have low precipitation often have low C content, and identify if these type of ecosystems would be really less vulnerable to increased precipitation frequency or not. 3) I would suggest the authors to expand their discussion on plant contribution. Indeed, as plant respiration (or better, rhizosphere respiration) covers, often, the greater proportion of C derived from soil, it would be interesting to discuss more in details how would plant respiration contribute to these effects. Also another thing that the authors could discuss in this context is that all the theoretical background and the parameters that they take into consideration emerge from either laboratory incubation (therefore no plant contribution at all), or in field experiment where they calculated the contribution of heterotrophic to total. However in both these cases it is completely neglected the heterotrophic contribution to respiration

of freshly produce plant compounds. I know this parameter is theoretically impossible to measure, but I think it is good to point it out. 4) Point number 3 could be further linked with a small discussion on the effects of changes in precipitation depending on the season. Indeed, as the magnitude of respiration depend on temperature (and also plant input) and also the rate at which soil dries, I guess we would expect to have different results in different seasons. This could be briefly discussed.

SPECIFIC COMMENTS: L.230-231 Could you provide the graph were you compare the respiration normalized by the amount of SOC?

L.388 observed by who? The author in the cited paper? Please specify.

---

## Author Comment (AC1) · 22 Apr 2020

**Response to Anonymous Referee #1**

We thank Referee #1 for his/her comments, which are addressed as explained below. The Referee's text is reported in Italic and our responses in roman.

*I reviewed the manuscript n. bg-2020-95 entitled "Rainfall intensification increases the contribution of rewetting pulses to soil respiration". In this manuscript authors assess the validity of a stochastic model for soil heterotrophic respiration to explain the effects of precipitation variability on soil $CO_2$ fluxes. The manuscript is very well written and the rationale for the need of this model is well formulated. The theoretical considerations are sound, and the equations look correct. Authors also address the limits of their model in the discussion section (which is important when the readers will try to link to the model to their own experiment or for general extrapolation) and tend to not emphasize too much the findings of their model given the discussed limitation. Nevertheless, the study is important because highlights an important aspect, which is the growing importance of rewetting events depending on precipitation distribution. This is an aspect that is often overlooked, especially in manuscript dealing with drought, and highlight the importance of context dependent effects. I have no reserves in recommending this paper for publication, with only minor revisions and suggestions from my side. This has happened to me rarely and therefore I must compliment the authors for the excellent work.*

We thank the reviewer for the encouraging words and the constructive comments, which are addressed in the following.

*My general comments are mainly small recommendation and suggestions:*

*1) I would recommend the authors to be more careful when they refer to soil respiration and to heterotrophic respiration. Indeed, in all their theoretical background, consideration is given only to microbial respired carbon and therefore neglecting the contribution of plant roots to drying and rewetting events. While this is fine for the paper, I would urge the authors to refer to heterotrophic soil respiration rather than to soil respiration mostly throughout the paper.*

We agree with the reviewer that only heterotrophic respiration is considered in our model framework, and indeed we used respiration data from incubations without plants to parameterize the model. To make sure that it is clear we only refer to heterotrophic respiration, we will include the word "heterotrophic" in key positions in the text—namely in section titles and in the first sentences of each section. We will also change the manuscript title to:

"Rainfall intensification increases the contribution of rewetting pulses to soil heterotrophic respiration"

Finally, to avoid ambiguity, we will explicitly refer to "total heterotrophic respiration" in (almost) all instances, as the word "total" alone might erroneously suggest "heterotrophic plus autotrophic soil respiration".

*2) I understand that for simplicity authors calculate average parameters between soil with high OC and low OC (L. 233-235) for further theoretical considerations. However, wouldn't be also very interesting to get parameters for both (high and low OC) and understand whether we can draw similar conclusions in both cases? Also, this would make it easier to then look at the larger picture, as you in part address (L. 356-358), that systems that have low precipitation often have*

*low C content, and identify if these types of ecosystems would be really less vulnerable to increased precipitation frequency or not.*

This is a good point—indeed changes in the maximum respiration rates during drying ($R_{d,max}$) and the maximum respiration pulse at rewetting ($R_{r,max}$) are expected because of different soil organic carbon contents. Large values of $R_{r,max}$ relative to $R_{d,max}$ will increase the contribution of respiration pulses. The (few) data points we have provide some indications of how $R_{r,max}$ and $R_{d,max}$ co-vary (Figure R1). An exponential regression explains most of the variability in the data and suggests that $R_{r,max}$ increases disproportionately with increasing $R_{d,max}$. This means that in soils with less organic matter (i.e., lower $R_{r,max}$ and $R_{d,max}$), the relative role of respiration pulses could be lower just because of the smaller $R_{r,max}/R_{d,max}$ ratio. It is possible that the nonlinear increase in respiration pulses is due to the disproportional increase in microbial biomass as soil organic matter increases—more microbes ready to be activated at rewetting will drive a larger respiration pulse. This can affect our conclusions regarding the sensitivity of C-rich soils to changes in precipitation patterns.

To test how different absolute values of and ratios between $R_{r,max}$ to $R_{d,max}$ alter the effect of rainfall statistical properties on the partitioning of heterotrophic respiration, we can produce the equivalent of Figures 5 and 6 in the original manuscript, but now using the mean $R_{r,max}$ and $R_{d,max}$ of mineral and organic soils separately. These new results are shown in Figures R2-R5. The first two figures (R2 and R3) respectively show the mean respiration rates obtained when parameterizing the model for mineral and organic soils. The last two figures (R4 and R5) show instead the standard deviations and coefficients of variation, again for mineral and organic soils, respectively.

[Figure]

**Figure R1. Co-variation of the maximum respiration rate during drying ($R_{d,max}$) and the maximum respiration pulse at rewetting ($R_{r,max}$). The curve represents the nonlinear least square regression of the points using an exponential function. Data points are from Table 2 in the manuscript.**

[Figure]

**Figure R2 (Mineral soils): Effect of precipitation statistical properties (mean event frequency $\lambda$ and depth $\alpha$) on the mean heterotrophic respiration rates during dry periods $\langle R_d \rangle$ (a), and at rewetting $\langle R_r^* \rangle$ (b), the mean total respiration rate $\langle R_t \rangle$ (c), and on the fraction of the total heterotrophic respiration rate due to rewetting pulses $\langle R_r^* \rangle / \langle R_t \rangle$ (d). The white contour curves indicate combinations of $\lambda$ and $\alpha$ that generate different annual precipitation rates ($\langle P \rangle = \alpha\lambda$ =0.25, 0.5, 1, and 2 m y$^{-1}$ from dotted to solid lines). Other parameter values: $R_{r,max}$=3.3 gC m$^{-2}$, $R_{d,max}$=0.8 gC m$^{-2}$ d$^{-1}$, $b$=0.12, $Z_r$=0.3 m, $n$=0.5, $s_w$=0.2, $s_1$=0.7, $E_{max}$=0.0037 m d$^{-1}$.**

[Figure]

**Figure R3 (Organic soils): Effect of precipitation statistical properties (mean event frequency $\lambda$ and depth $\alpha$) on the mean heterotrophic respiration rates during dry periods $\langle R_d \rangle$ (a), and at rewetting $\langle R_r^* \rangle$ (b), the mean total respiration rate $\langle R_t \rangle$ (c), and on the fraction of the total heterotrophic respiration rate due to rewetting pulses $\langle R_r^* \rangle / \langle R_t \rangle$ (d). The white contour curves indicate combinations of $\lambda$ and $\alpha$ that generate different annual precipitation rates ($\langle P \rangle = \alpha \lambda$ =0.25, 0.5, 1, and 2 m y$^{-1}$ from dotted to solid lines). Other parameter values: $R_{r,max}$=14.6 gC m$^{-2}$, $R_{d,max}$=1.3 gC m$^{-2}$ d$^{-1}$, $b$=0.11, $Z_r$=0.3 m, $n$=0.5, $s_w$=0.2, $s_1$=0.7, $E_{max}$=0.0037 m d$^{-1}$.**

[Figure]

**Figure R4 (Mineral soils): Effect of precipitation statistical properties (mean event frequency $\lambda$ and depth $\alpha$) on the standard deviations of heterotrophic respiration rates during dry periods $\sigma_{R_d}$ (a) and respiration pulses at rewetting $\sigma_{R_r}$ (b), and on the coefficients of variations of respiration rates during dry periods $CV_{R_d}$ (c) and respiration pulses at rewetting $CV_{R_r}$ (d). The white contour curves indicate combinations of $\lambda$ and $\alpha$ that generate different annual precipitation rates ($\langle P \rangle = \alpha\lambda$ =0.25, 0.5, 1, and 2 m y$^{-1}$ from dotted to solid lines). Other parameter values are as in Figure R2.**

[Figure]

**Figure R5 (Organic soils): Effect of precipitation statistical properties (mean event frequency $\lambda$ and depth $\alpha$) on the standard deviations of heterotrophic respiration rates during dry periods $\sigma_{R_d}$ (a) and respiration pulses at rewetting $\sigma_{R_r}$ (b), and on the coefficients of variations of respiration rates during dry periods $CV_{R_d}$ (c) and respiration pulses at rewetting $CV_{R_r}$ (d). The white contour curves indicate combinations of $\lambda$ and $\alpha$ that generate different annual precipitation rates ($\langle P \rangle = \alpha\lambda$ =0.25, 0.5, 1, and 2 m y$^{-1}$ from dotted to solid lines). Other parameter values are as in Figure R3.**

Figures R2-R5 show that in general patterns are conserved across soils with different soil organic C content. In particular, comparing Figures R2d and R3d suggests that the increasing contribution of respiration pulses with rainfall intensification holds regardless of soil organic C content, but the absolute values differ—in organic soils (Figure R3), the contribution of respiration pulses is consistently higher than in mineral soils. This is expected given the different values of $R_{r,max}$ (note that values of $R_{d,max}$ are more comparable across soils). Moreover, Figures R4c-d and R5c-d confirm our expectation (stated at the end of Section 3.4 of the manuscript) that patterns in the coefficients of variation are conserved more than patterns in standard deviation.

To summarize the content of Figures R2-R5, we can plot the fraction of total heterotrophic respiration contributed by respiration pulses as a function of rainfall event frequency, for set values of total precipitation (i.e., as the frequency increases, the mean event depth decreases), and for both mineral and organic soils. The result is shown in Figure R6. If offered the opportunity to revise the manuscript, we will add the mean values of the respiration model parameters for mineral and organic soils in Table 2, include Figure R6 in the manuscript (as Figure 7), and expand the Results with a new sub-section to present the new figure:

"**3.5 Effects of rainfall intensification and organic C availability on heterotrophic respiration**
Results shown in Figures 5 and 6 are based on average respiration model parameters; here, we explore how changing organic C content from mineral to organic soils affects the contribution of rewetting pulses to total soil heterotrophic respiration. We also focus on changes in respiration patterns along gradients of rainfall intensification; i.e., decreasing precipitation frequency $\lambda$ while precipitation event depth $\alpha$ is increased and total precipitation is kept fixed (as along the white contour curves in Figures 5 and 6). Figure 7 shows that rainfall intensification decreases $\langle R_t \rangle$ (Figure 7a), but increases $\langle R_r^* \rangle / \langle R_t \rangle$ (Figure 7b), regardless of soil organic C availability (black vs. grey curves) and total precipitation (dashed vs. solid curves). However, for a given total precipitation, organic soils (grey curves) exhibit both higher $\langle R_t \rangle$ and higher $\langle R_r^* \rangle / \langle R_t \rangle$ than mineral soils (black curves), due to their higher $R_{r,max}$ (Table 2). Moreover, in both soils, higher precipitation increases $\langle R_t \rangle$ while decreasing $\langle R_r^* \rangle / \langle R_t \rangle$ (compare solid vs. dashed curves)."

[Figure]

**Figure R6: (a) Total heterotrophic respiration $\langle R_t \rangle$, and (b) fraction of the total heterotrophic respiration rate due to rewetting pulses, $\langle R_r^* \rangle / \langle R_t \rangle$, as a function of mean precipitation event frequency $\lambda$, for given total precipitation (i.e., $\alpha = \lambda / < P >$), and for both mineral and organic soils. The respiration model parameters are reported in Table 2 and other parameter values are as in Figure R2.**

*3) I would suggest the authors to expand their discussion on plant contribution. Indeed, as plant respiration (or better, rhizosphere respiration) covers, often, the greater proportion of C derived*

*from soil, it would be interesting to discuss more in details how would plant respiration contribute to these effects. Also, another thing that the authors could discuss in this context is that all the theoretical background and the parameters that they take into consideration emerge from either laboratory incubation (therefore no plant contribution at all), or in field experiment where they calculated the contribution of heterotrophic to total. However, in both these cases it is completely neglected the heterotrophic contribution to respiration of freshly produce plant compounds. I know this parameter is theoretically impossible to measure, but I think it is good to point it out.*

We agree that plant contribution to total soil respiration is important (autotrophic respiration), and that the model is parameterized using data from experiments without plants (thus neglecting heterotrophic respiration stemming from fresh C inputs from roots in the rhizosphere). In the submitted manuscript, we hint at the autotrophic contribution in Discussion Section 4.3, but we can further expand the Discussion on these topics as suggested, by adding in Section 4.2:

"Our focus in this contribution is on heterotrophic respiration, but the data we used to parameterize the model are from laboratory studies without plants. Therefore, our heterotrophic respiration estimates neglect contributions from fresh C inputs from roots to the rhizosphere, which are known to be important (Finzi et al., 2015; Kuzyakov and Gavrichkova, 2010).. However, the timing of rhizodeposition depends on plant activity, which in turn depends on previous environmental conditions—differently from soil microbes that respond to soil moisture changes rapidly, plant responses integrate previous conditions thereby partly decoupling root activity from current soil moisture. It is thus non-trivial to include rhizosphere processes in the current framework."

For the sake of space, and given the focus on heterotrophic respiration, we would prefer not to expand the Discussion on autotrophic contributions to total soil respiration beyond the points already presented in the existing Section 4.3.

*4) Point number 3 could be further linked with a small discussion on the effects of changes in precipitation depending on the season. Indeed, as the magnitude of respiration depend on temperature (and also plant input) and also the rate at which soil dries, I guess we would expect to have different results in different seasons. This could be briefly discussed.*

This is also a good point. The proposed model assumes statistical steady state, and can be applied in the growing season of water-limited ecosystems, when temperature is not the main limiting factor. To address this comment, we can add a paragraph at the end of Section 4.3:

"Our findings are based on time-invariant relations between heterotrophic respiration and soil moisture, but temperature and other environmental conditions also affect microbial activity—in part directly and in part indirectly via rhizodeposition—raising the question of how our results could be impacted by other respiration controlling factors. As a first approximation, temperature could be assumed to alter directly both respiration rates during drying and respiration pulses in similar way. This implies that our results would hold even under fluctuating temperatures, at least during the growing season, when temperature variations are limited and precipitation can be described by a simple marked Poisson process (Section 2.1.1). However, a different modelling approach would be needed to quantify the mean heterotrophic respiration rate during seasons with frequent rainfall events, when respiration pulses are likely to be less important and anaerobic conditions (here neglected) could play a role. As the time scale expands from the growing season to the whole year, also seasonal fluctuations in plant activity that delay the supply of C substrates to microbes will play a role (Finzi et al., 2015; Kuzyakov and Gavrichkova, 2010), leading to a

hierarchy of responses at multiple time scales—a more complex problem than the one addressed in this contribution."

*L.230-231 Could you provide the graph where you compare the respiration normalized by the amount of SOC?*

This is a good idea, as this representation allows for a simpler description of the fitting of the respiration pulse model. Figure R7 shows the same data and model as in Figure 2 in the original manuscript, but now normalizing the respiration pulse size by the soil organic carbon concentration in the soil sample. Indeed, results are more comparable across soil types. Following this suggestion, we will change current Figure 2 with Figure R7, and revise the description of this figure in the Results accordingly:

"In Figure 2, respiration pulses at rewetting are normalized by the amount of organic C in each soil to facilitate comparisons. However, after accounting for variations in organic C content, bulk density and soil layer depth, $R_{r,max}$ and $R_{d,max}$ values on a per unit ground area basis are higher in the organic soils than in mineral soils (Table 2), and so is the ratio between $R_{r,max}$ and $R_{d,max}$."

[Figure]

**Figure R7: Relations between respiration pulse size ($R_r$, normalized by soil organic C content) and pre-wetting soil moisture ($x_d$) and soil moisture increment at rewetting ($y$), for six soils; top row: mineral soils; bottom row: organic soils (note different vertical scales between rows). Symbols represent measured respiration pulses and surfaces are fitted $R_r$ functions from Eq. (13)**Error! Reference source not found. **(soil characteristics, fitting parameters, and data sources are reported in Table 2).**

*L.388 observed by who? The author in the cited paper? Please specify.*

We were referring to the study by Harper et al. (2005) cited in the same sentence. Since the second clause in that sentence is our conclusion, we prefer to keep the citation at the end of the first clause. However, we can re-phrase for clarity:

"Our approach neglects the lower plant C inputs and contributions to total soil respiration under a more intermittent precipitation regime (Harper et al., 2005), which further reduces the total (combined autotrophic and heterotrophic) soil respiration rate."

**References**

Finzi, A. C., Abramoff, R. Z., Spiller, K. S., Brzostek, E. R., Darby, B. A., Kramer, M. A. and Phillips, R. P.: Rhizosphere processes are quantitatively important components of terrestrial carbon and nutrient cycles, Glob. Change Biol., 21(5), 2082–2094, doi:10.1111/gcb.12816, 2015.

Harper, C. W., Blair, J. M., Fay, P. A., Knapp, A. K. and Carlisle, J. D.: Increased rainfall variability and reduced rainfall amount decreases soil CO2 flux in a grassland ecosystem, Glob. Change Biol., 11(2), 322–334, 2005.

Kuzyakov, Y. and Gavrichkova, O.: REVIEW: Time lag between photosynthesis and carbon dioxide efflux from soil: a review of mechanisms and controls, Glob. Change Biol., 16(12), 3386–3406, doi:10.1111/j.1365-2486.2010.02179.x, 2010.

---

## Referee Comment (RC2) · Anonymous Referee #2 · 2 Jun 2020

In this manuscript, the authors present an analysis of the role of variability in rainfall (amount and frequency) on average heterotrophic respiration for dry ecosystems. It is an attempt to provide a mechanistic explanation of the 'Birch' effect, showing that respiration pulses are more relevant under low precipitation frequency and to a lesser extent to low rainfall amounts. The analysis relies on a very simple stochastic model that links a stochastic representation of rainfall to changes in moisture and subsequent changes in heterotrophic respiration. The resulting probability density functions (PDFs) are then expressed in terms of their first and second moments. I appreciate the effort of the authors in highlighting the limitations of the model ,because due to its simplicity, it excludes a number of process commonly integrated in soil respiration models. However, I agree with the authors in that the minimal model presented here is enough to

capture the main dominant processes that capture the stochastic nature of respiration pulses. This is enough to provide a general understanding of the problem, but these results should be used with caution when attempting to represent other type of systems; e.g. systems with high seasonality in litter inputs or water-logged. I have only very minor comments, and can recommend the manuscript for publication after small modifications.

Minor comments:

- Line 100. The parameter $\gamma$ is an important parameter because it controls the exponential decay in both terms of equation (3). Please give a more intuitive explanation of this parameter.

- Line 105. I think the term 'stochastic steady-state' is not appropriate. A steady-state refers to a fixed point in the phase plane, which is not the case for the stochastic case here. I suggest using the term 'stationary' instead.

- Eq. (8). I think it would be helpful for the reader to split this equation in two lines, with the first line using an inverse function notation, so it is clear that your aim is to find the inverse of equation (7). Also, I think it's important to explicitly mention that finding this inverse implies that you can only use this approach for bijective functions, and not for non-bijective functions. There are many functions in the literature that show that respiration as a function of soil moisture has a maximum at an intermediate level, after which respiration declines with moisture (e.g. Skopp 1990, SSSAJ 54:1619). These functions are non-bijective and therefore have no inverse over their entire domain. A consequence of this, which I think is partially addressed by the authors, is that you can only use this approach when predicting the behavior of respiration at low soil moisture levels.

---

## Author Comment (AC2) · 5 Jun 2020

**Response to Anonymous Referee #2**

We thank Referee #2 for his/her comments, which are addressed as explained below. The Referee's text is reported in Italic and our responses in roman.

*In this manuscript, the authors present an analysis of the role of variability in rainfall (amount and frequency) on average heterotrophic respiration for dry ecosystems. It is an attempt to provide a mechanistic explanation of the 'Birch' effect, showing that respiration pulses are more relevant under low precipitation frequency and to a lesser extent to low rainfall amounts. The analysis relies on a very simple stochastic model that links a stochastic representation of rainfall to changes in moisture and subsequent changes in heterotrophic respiration. The resulting probability density functions (PDFs) are then expressed in terms of their first and second moments. I appreciate the effort of the authors in highlighting the limitations of the model, because due to its simplicity, it excludes a number of process commonly integrated in soil respiration models. However, I agree with the authors in that the minimal model presented here is enough to capture the main dominant processes that capture the stochastic nature of respiration pulses. This is enough to provide a general understanding of the problem, but these results should be used with caution when attempting to represent other type of systems; e.g. systems with high seasonality in litter inputs or water-logged. I have only very minor comments, and can recommend the manuscript for publication after small modifications.*

The reviewer nicely summarized the philosophy of our contribution—indeed, we do not aim to describe in a fully mechanistic way all processes involved, but to analytically link the statistical properties of heterotrophic respiration to those of precipitation. This approach represents also the main novelty of this contribution. To further emphasize model limitations along the lines suggested by the reviewer, we can add a further sentence in the Discussion:

"In addition to these limitations, our results should also be interpreted with caution when rainfall seasonality is important, because the assumption of stochastic stationarity may not be met, requiring the derivation of a different probability density function of soil moisture (e.g., Vico et al., 2017). Nevertheless, our results will still hold for parts of the year when the rainfall regime is relatively stable."

*Minor comments:*
*• Line 100. The parameter gamma is an important parameter because it controls the exponential decay in both terms of equation (3). Please give a more intuitive explanation of this parameter.*

We can include the explanation: "$\gamma$ can be interpreted as the number of average rainfall events needed to replenish the plant available soil water."

*• Line 105. I think the term 'stochastic steady-state' is not appropriate. A steadystate refers to a fixed point in the phase plane, which is not the case for the stochastic case here. I suggest using the term 'stationary' instead.*

Following this suggestion, the term "steady state" will be changed to "stationary" throughout the manuscript.

*• Eq. (8). I think it would be helpful for the reader to split this equation in two lines, with the first line using an inverse function notation, so it is clear that your aim is to find the inverse of equation (7).*

This would be a good idea had we defined a function $R_d = f(x)$. Then the inverse $x = f^{-1}(R_d)$ could be used in Eq. 8. We have not defined such a function and more simply expressed respiration as a function of soil moisture in Eq. 7. We would prefer not to add a further symbol $f$ here and leave the notation as is. However, we can add a clarification:

"… the PDF of soil moisture is evaluated at moisture values corresponding to given respiration values. This is done by inverting Eq. (7) and expressing $x$ as a function of $R_d$"

*Also, I think it's important to explicitly mention that finding this inverse implies that you can only use this approach for bijective functions, and not for non-bijective functions. There are many functions in the literature that show that respiration as a function of soil moisture has a maximum at an intermediate level, after which respiration declines with moisture (e.g. Skopp 1990, SSSAJ 54:1619). These functions are non-bijective and therefore have no inverse over their entire domain. A consequence of this, which I think is partially addressed by the authors, is that you can only use this approach when predicting the behavior of respiration at low soil moisture levels.*

The reviewer is correct that a non-monotonic (and thus non-bijective) function would require considering separately the different domains of soil moisture in which the respiration function is (locally) monotonic. To clarify this issue, we will add the following sentences:

"We note that Eq. (7) is monotonic in the domain $0 \leq x \leq 1$, which allows defining unambiguously the inverse of $R_d(x)$. Had we used a non-monotonic $R_d(x)$ function (e.g., for applications of this approach to soils experiencing long saturation periods), the derived distribution approach would have required splitting the $x$ domain into two—one for each monotonic branch of $R_d(x)$."

**References**

Vico, G., Dralle, D., Feng, X., Thompson, S. and Manzoni, S.: How competitive is drought deciduousness in tropical forests? A combined eco-hydrological and eco-evolutionary approach, Environ. Res. Lett., 12(6), 065006, doi:10.1088/1748-9326/aa6f1b, 2017.

---

## Author Response (AR1)

**Response to the Editor**

We thank Dr. Hagedorn for encouraging us to submit a revised manuscript. His comments are addressed in the following. We have already detailed how we planned to revise our manuscript in the responses to the reviewers' comments posted in the public discussion forum. In this letter, we only describe how the planned changes have been implemented, and leave our arguments in support to those changes in the open discussion to avoid unnecessary repetitions. All comments by the Editor and the Referees are reported in Italic and our responses are in roman.

*Both reviewers found that this is a great manuscript on the impact of rainfall variability and rewetting on heterotrophic respiration. It reads very well and it is a great model exploration of drying and rewetting effects which was so far lacking. Congratulations!*

Thanks for the supporting comments.

*Three very minor comments from my side:*
1) *make clear to consistently use heterotrophic respiration instead of soil respiration as soil respiration is generally used to comprise both auto- and heterotrophic respiration.*

Throughout the manuscript, we now refer explicitly to "heterotrophic respiration", except in a few instances in the Discussion, where we write about "total soil respiration", where we mean to include both autotrophic and heterotrophic components. Also the title was updated along these lines to "Rainfall intensification increases the contribution of rewetting pulses to soil heterotrophic respiration"

2) *write out abbreviations in Table 2 to make it self-understandable*

The table caption was amended to explain the symbols:

"Table 2: Characteristics of the selected mineral and organic soil samples; estimates of the respiration model parameters in Eq. (13) ($R_{d,max}$: maximum respiration rate at the soil field capacity, $R_{r,max}$: maximum respired carbon at rewetting) and coefficients of determination ($R^2$) for the least square fit of the data (see also Figure 2)."

3) *'accumulated DOC' (L. 47): How can DOC (=dissolved organic carbon) accumulate in dry soil? Either rename it to extractable OC (this is what is usually measured), potentially leachable DOC or mobilizable OC.*

In principle, it is possible that the amount of C (mass per unit ground area) in dissolved form accumulates in drying soil, because the dissolved C concentration (mass per unit soil water volume) increases faster than the rate of decrease of soil water content. However, we agree that the used terminology can create ambiguities and changed the text as suggested.

In addition, we streamlined the text to improve flow and updated the citations with several recent articles (Barnard et al., 2020; Lopez-Ballesteros et al., 2016; Tang et al., 2019).

**Response to Anonymous Referee #1**

*I reviewed the manuscript n. bg-2020-95 entitled "Rainfall intensification increases the contribution of rewetting pulses to soil respiration". In this manuscript authors assess the validity of a stochastic model for soil heterotrophic respiration to explain the effects of precipitation variability on soil CO2 fluxes. The manuscript is very well written and the rationale for the need of this model is well formulated. The theoretical considerations are sound, and the equations look correct. Authors also address the limits of their model in the discussion section (which is important when the readers will try to link to the model to*

*their own experiment or for general extrapolation) and tend to not emphasize too much the findings of their model given the discussed limitation. Nevertheless, the study is important because highlights an important aspect, which is the growing importance of rewetting events depending on precipitation distribution. This is an aspect that is often overlooked, especially in manuscript dealing with drought, and highlight the importance of context dependent effects. I have no reserves in recommending this paper for publication, with only minor revisions and suggestions from my side. This has happened to me rarely and therefore I must compliment the authors for the excellent work.*

We thank the reviewer for the encouraging words and the constructive comments, which are addressed in the following.

*My general comments are mainly small recommendation and suggestions:*

*1) I would recommend the authors to be more careful when they refer to soil respiration and to heterotrophic respiration. Indeed, in all their theoretical background, consideration is given only to microbial respired carbon and therefore neglecting the contribution of plant roots to drying and rewetting events. While this is fine for the paper, I would urge the authors to refer to heterotrophic soil respiration rather than to soil respiration mostly throughout the paper.*

To make sure that it is clear we only refer to heterotrophic respiration, we included the word "heterotrophic" in key positions in the text—namely in section titles and in the first sentences of each section. We also changed the manuscript title to: "Rainfall intensification increases the contribution of rewetting pulses to soil heterotrophic respiration"

Finally, to avoid ambiguity, we explicitly refer to "total heterotrophic respiration" in (almost—except a case in the Discussion) all instances, as the word "total" alone might erroneously suggest "heterotrophic plus autotrophic soil respiration".

*2) I understand that for simplicity authors calculate average parameters between soil with high OC and low OC (L. 233-235) for further theoretical considerations. However, wouldn't be also very interesting to get parameters for both (high and low OC) and understand whether we can draw similar conclusions in both cases? Also, this would make it easier to then look at the larger picture, as you in part address (L. 356-358), that systems that have low precipitation often have low C content, and identify if these types of ecosystems would be really less vulnerable to increased precipitation frequency or not.*

We fundamentally agree and have presented our arguments in the response to the reviewer's comments posted in the public discussion forum. In the revised manuscript, we present a new figure where the fraction of total heterotrophic respiration contributed by respiration pulses is shown as a function of rainfall event frequency, for set values of total precipitation (i.e., as the frequency increases, the mean event depth decreases), and for both mineral and organic soils (Figure 7, shown below for convenience).

[Figure]

**Figure 7: (a) Total heterotrophic respiration $\langle R_t \rangle$, and (b) fraction of the total heterotrophic respiration rate due to rewetting pulses, $\langle R_r^* \rangle / \langle R_t \rangle$, as a function of mean precipitation event frequency $\lambda$, for given total precipitation (i.e., $\alpha = \lambda/\langle P \rangle$), and for both mineral and organic soils. The respiration model parameters are reported in Table 2 and other parameter values are as in Figure 5.**

The new Figure 7 is described in the Results with a new sub-section:

**"3.5 Effects of rainfall intensification and organic C availability on heterotrophic respiration**
Results shown in Figures 5 and 6 are based on average respiration model parameters; here, we explore how changing organic C content from mineral to organic soils affects the contribution of rewetting pulses to total soil heterotrophic respiration. We also focus on changes in respiration patterns along gradients of rainfall intensification; i.e., decreasing precipitation frequency $\lambda$ while precipitation event depth $\alpha$ is increased and total precipitation is kept fixed (as along the white contour curves in Figures 5 and 6). Figure 7 shows that rainfall intensification decreases $\langle R_t \rangle$ (Figure 7a), but increases $\langle R_r^* \rangle / \langle R_t \rangle$ (Figure 7b), regardless of soil organic C availability (black vs. grey curves) and total precipitation (dashed vs. solid curves). However, for a given total precipitation, organic soils (grey curves) exhibit both higher $\langle R_t \rangle$ and higher $\langle R_r^* \rangle / \langle R_t \rangle$ than mineral soils (black curves), due to their higher $R_{r,max}$ (Table 2). As a result, in organic soils, the contribution of respiration pulses can be as high as 20% of the total heterotrophic respiration, whereas in mineral soils it tends to be lower than 10%. Moreover, in both soils, higher precipitation increases $\langle R_t \rangle$ while decreasing $\langle R_r^* \rangle / \langle R_t \rangle$ (compare solid *vs.* dashed curves)."

We also highlight these results in the abstract:

"Our results suggest that higher rainfall intermittency at constant total rainfall can increase the contribution of respiration pulses up to ~10 or 20% of the total heterotrophic respiration in mineral and organic soils, respectively."

*3) I would suggest the authors to expand their discussion on plant contribution. Indeed, as plant respiration (or better, rhizosphere respiration) covers, often, the greater proportion of C derived from soil, it would be interesting to discuss more in details how would plant respiration contribute to these effects. Also, another thing that the authors could discuss in this context is that all the theoretical background and the parameters that they take into consideration emerge from either laboratory incubation (therefore no plant contribution at all), or in field experiment where they calculated the contribution of heterotrophic to total. However, in both these cases it is completely neglected the heterotrophic contribution to respiration of freshly produce plant compounds. I know this parameter is theoretically impossible to measure, but I think it is good to point it out.*

In the revised manuscript, we have expanded the Discussion as suggested, by adding in Section 4.2:

"Our focus in this contribution is on heterotrophic respiration, but the data we used to parameterize the model are from laboratory studies without plants. Therefore, our heterotrophic respiration estimates neglect contributions from fresh C inputs from roots to the rhizosphere, which are known to be important (Finzi et al., 2015; Kuzyakov and Gavrichkova, 2010). However, the timing of rhizodeposition depends on plant activity, which in turn depends on previous environmental conditions—differently from soil microbes that respond to soil moisture changes rapidly, plant responses integrate previous conditions thereby partly decoupling root activity from current soil moisture. It is thus non-trivial to include rhizosphere processes in the current framework."

For the sake of space, and given the focus on heterotrophic respiration, we would prefer not to expand the Discussion on autotrophic contributions to total soil respiration beyond the points already presented in the existing Section 4.3.

*4) Point number 3 could be further linked with a small discussion on the effects of changes in precipitation depending on the season. Indeed, as the magnitude of respiration depend on temperature (and also plant input) and also the rate at which soil dries, I guess we would expect to have different results in different seasons. This could be briefly discussed.*

To address this comment, we have added a paragraph at the end of Section 4.3:

"Our findings are based on time-invariant relations between heterotrophic respiration and soil moisture, but temperature and other environmental conditions also affect microbial activity—in part directly and in

part indirectly via rhizodeposition—raising the question of how our results could be impacted by other respiration controlling factors. As a first approximation, temperature could be assumed to alter directly both respiration rates during drying and respiration pulses in similar way. This implies that our results would hold even under fluctuating temperatures, at least during the growing season, when temperature variations are limited and precipitation can be described by a simple marked Poisson process (Section 2.1.1). However, a different modelling approach would be needed to quantify the mean heterotrophic respiration rate during seasons with frequent rainfall events, when respiration pulses are likely to be less important and anaerobic conditions (here neglected) could play a role. As the time scale expands from the growing season to the whole year, also seasonal fluctuations in plant activity that delay the supply of C substrates to microbes will play a role (Finzi et al., 2015; Kuzyakov and Gavrichkova, 2010), leading to a hierarchy of responses at multiple time scales—a more complex problem than the one addressed in this contribution."

*L.230-231 Could you provide the graph where you compare the respiration normalized by the amount of SOC?*

Following this suggestion, we have modified Figure 2 (now showing respiration rates normalized by the amount of SOC) and revised the description of this figure in the Results accordingly:

"In Figure 2, respiration pulses at rewetting are normalized by the amount of organic C in each soil to facilitate comparisons. However, after accounting for variations in organic C content, bulk density and soil layer depth, values of $R_{r,max}$ and $R_{d,max}$ per unit ground area are higher in the organic soils than in mineral soils (Table 2), and so is the ratio between $R_{r,max}$ and $R_{d,max}$."

[Figure]

**Figure 2: Relations between respiration pulse size ($R_r$, normalized by soil organic C content) and pre-wetting soil moisture ($x_d$) and soil moisture increment at rewetting ($y$), for six soils; top row: mineral soils; bottom row: organic soils (note different vertical scales between rows). Symbols represent measured respiration pulses and surfaces are fitted $R_r$ functions from Eq. (13) (soil characteristics, fitting parameters, and data sources are reported in Table 2).**

*L.388 observed by who? The author in the cited paper? Please specify.*

We have re-phrased for clarity:

[revised manuscript text omitted]